# Biomarkers Associated with Organ-Specific Involvement in Juvenile Systemic Lupus Erythematosus

**DOI:** 10.3390/ijms22147619

**Published:** 2021-07-16

**Authors:** James Greenan-Barrett, Georgia Doolan, Devina Shah, Simrun Virdee, George A. Robinson, Varvara Choida, Nataliya Gak, Nina de Gruijter, Elizabeth Rosser, Muthana Al-Obaidi, Maria Leandro, Michael S. Zandi, Ruth J. Pepper, Alan Salama, Elizabeth C. Jury, Coziana Ciurtin

**Affiliations:** 1Centre for Adolescent Rheumatology Versus Arthritis, University College London, London WC1E 6DH, UK; james.greenan@nhs.net (J.G.-B.); georgia.doolan.17@ucl.ac.uk (G.D.); devina.shah@nhs.net (D.S.); george.robinson.15@ucl.ac.uk (G.A.R.); v.choida@ucl.ac.uk (V.C.); n.gruijter@ucl.ac.uk (N.d.G.); e.rosser@ucl.ac.uk (E.R.); 2Department of Ophthalmology, Royal Free Hospital, London NW3 2QG, UK; zchassv@ucl.ac.uk; 3Department of Rheumatology, University College London Hospital NHS Foundation Trust, London NW1 2BU, UK; nataliya.gak2@nhs.net (N.G.); maria.leandro@ucl.ac.uk (M.L.); 4Department of Paediatric Rheumatology, Great Ormond Street Hospital, London WC1N 3JH, UK; Muthana.AlObaidi@gosh.nhs.uk; 5NIHR Biomedical Research Centre, UCL Great Ormond Street Institute of Child Health, London WC1N 1EH, UK; 6Centre for Rheumatology, Division of Medicine, University College London, London WC1E 6DH, UK; e.jury@ucl.ac.uk; 7Department of Neurology, National Hospital for Neurology and Neurosurgery, University College London Hospitals NHS Foundation Trust, London NW1 2BU, UK; michael.zandi@nhs.net; 8Department of Renal Medicine, Royal Free Hospital, University College London, London NW3 2QG, UK; ruth.pepper@nhs.net (R.J.P.); a.salama@ucl.ac.uk (A.S.)

**Keywords:** juvenile systemic lupus erythematosus, organ-specific biomarkers, renal biomarkers, cardiovascular biomarkers, haematological biomarkers, neurological biomarkers, skin disease biomarkers, arthritis biomarkers, genetic biomarkers

## Abstract

Juvenile systemic lupus erythematosus (JSLE) is characterised by onset before 18 years of age and more severe disease phenotype, increased morbidity and mortality compared to adult-onset SLE. Management strategies in JSLE rely heavily on evidence derived from adult-onset SLE studies; therefore, identifying biomarkers associated with the disease pathogenesis and reflecting particularities of JSLE clinical phenotype holds promise for better patient management and improved outcomes. This narrative review summarises the evidence related to various traditional and novel biomarkers that have shown a promising role in identifying and predicting specific organ involvement in JSLE and appraises the evidence regarding their clinical utility, focusing in particular on renal biomarkers, while also emphasising the research into cardiovascular, haematological, neurological, skin and joint disease-related JSLE biomarkers, as well as genetic biomarkers with potential clinical applications.

## 1. Introduction

The research progress achieved in the last decades in rheumatology led to opportunities for discovery and testing of various biomarkers for early detection, prognostic and response to treatment in autoimmune rheumatic diseases (ARDs), in a quest for more individualised and targeted management strategies. Systemic lupus erythematous (SLE) is the prototypical immune-complex medicated ARD, characterised by a chronic inflammatory process involving autoantibody production against self-antigens [1], leading to various organs and systems involvement. When diagnosed before the age of 18 years [1,2,3,4], the disease is called juvenile SLE (JSLE) or SLE with childhood-onset and accounts for approximately 15–20% of all SLE patients [3,4,5]. The severity and clinical phenotype of SLE can vary considerably with an unpredictable disease course that is frequently relapsing and remitting in nature [3]. It is well recognised that JSLE is associated with a higher disease burden at diagnosis and a more aggressive clinical course when compared to adult-onset SLE, and thus it often associated with a poorer prognosis [1,2,3,5,6]. Major organ involvement occurs more frequently in JSLE compared to adult-onset SLE, and includes renal, neuropsychological, haematological, mucocutaneous, and cardiopulmonary manifestations, which in turn contributes to a significant increase in morbidity and mortality [3,7]. Lupus nephritis (LN) has increased prevalence in children with JSLE [6] compared to adult-onset SLE, and it is one of the main determining factors of poor prognosis [8], suggesting the need for better research focused on this patient population instead of extrapolating management strategies from adult-onset SLE.

The 2017 SHARE initiative (Single Hub and Access point for Paediatric Rheumatology in Europe) produced evidenced-based consensus recommendations for the diagnosis and treatment of JSLE, as well as assessed the sensitivity of classification criteria used in adult-onset SLE [9]. Despite recent efforts of improving the disease classification in adults, the American College of Rheumatology (ACR)/European League Against Rheumatism (EULAR) 2019 criteria did not perform better than the previous criteria (Systemic Lupus International Collaborating Clinics—SLICC 2012 and ACR 1997), possibly because of higher prevalence of ANA positivity in younger age, suggesting that the paediatric rheumatologist expertise is key in the process of evaluation of future classification efforts, especially if they aim to be implemented across age [10].

In addition to recognising the differences in clinical presentation and classification between JSLE and adult-onset SLE, clinicians face the challenge of implementing therapeutic strategies with limited evidence of efficacy in JSLE. The treatment of SLE across age is incredibly complex and reliant on multidisciplinary assessment in selected cases. There is a clear lack of good quality evidence, particularly derived from randomised control trials (RCTs) to support targeted JSLE treatments [5,9]. Therapeutic strategies in JSLE are therefore unstandardised and still led by expert opinion and primarily based on evidence from adult studies [5], despite a recent effort to include more children in RCTs in SLE. JSLE patients often require higher doses and longer courses of immunosuppressive treatment than patients with adult-onset SLE, and therefore are exposed to a greater risk of drug-related harm [4,9,11]. Because of the more severe clinical phenotype, less validated treatment strategies and increased morbidity and potential drug-related toxicity overall [12], further research to enable better characterisation of this patient population using markers which can inform personalised treatment approaches is required [13]. As JSLE is a life-long, potentially devastating disease, biomarkers have the potential to facilitate earlier diagnosis, predict response to treatment, monitor disease course, and aid the development of targeted treatments, which could ultimately help improve the prognosis in this cohort of patients.

Biomarkers in general reflect various biologic processes and comprise genomic, transcriptomic, molecular, histological, imaging, as well as clinical, physiological and routine serological/urine markers with different degrees of correlation with the disease manifestations or pathogenesis [14,15]. Identifying promising paediatric biomarkers is notably difficult, and it is of particular importance that they are applicable to paediatric specific pathology, and that their measurements are adjusted to age-dependent changes in physiology [16,17]. It is also desirable that the methods used for biomarker measurements are non-invasive as much as possible, and easily reproducible, bearing minimal potential complications and distress for the young patients [17,18]. If they fulfil meticulous testing criteria, some of the biomarkers can be validated as surrogate markers, and used to accurately substitute a clinical endpoint in an efficacy trial or clinical practice [18,19]. There is a pressing need to identify biomarkers that can be used as surrogate markers in JSLE to improve diagnosis and monitoring, as well as predict response to treatment or relapses, and support the development of targeted drug therapies [20]. 

Despite more advances achieved in SLE research in adults, there is a paucity of biomarkers and surrogate markers that have been validated or implemented in clinical practice. This could partly be explained by the complexity and heterogeneity of SLE, as well as a lack of full understanding of disease pathogenesis and its variable manifestations due to factors including age, ethnicity, and environmental influences [21,22]. The biomarkers tested in JSLE are often deduced from adult studies and are therefore of limited use, because of age-related differences in disease pathophysiology [16].

The aim of this review is to analyse the available literature related to traditional and novel biomarkers that have shown a promising role in identifying and predicting specific organ involvement in JSLE and appraise the evidence regarding their clinical utility. 

## 2. Methodology

### 2.1. Search Strategy

This is a narrative review based on literature searches (PubMed from January 1984 to December 2020) for systematic reviews, reviews, observational and interventional studies of patients with JSLE measuring various biomarkers and reporting on their clinical relevance, using the following MeSH terms: juvenile lupus OR lupus with childhood OR juvenile OR paediatric onset OR juvenile systemic lupus erythematosus OR lupus in paediatric population AND biomarkers AND lupus nephritis (renal biomarkers) OR cardiovascular disease (cardiovascular risk biomarkers) OR neuropsychiatric systemic lupus erythematosus (central nervous system biomarkers) OR arthritis (musculoskeletal biomarkers) OR cutaneous lupus erythematosus (skin biomarkers) OR serositis OR haematological biomarkers. Papers published in another language than English were excluded.

### 2.2. Study Selection

When selecting studies on renal biomarkers, studies on adult-onset SLE were excluded. However, due to the paucity of research on biomarkers in extra-renal JSLE, studies focused on adult-onset SLE were occasionally discussed. Case reports, case series, editorials and studies with low numbers of patients were excluded. This review is not intended to be exhaustive but rather focused on highlighting the relevance of various biomarkers that have been tested, and some validated, in JSLE with various organ involvement, proposing further research strategies to optimise the use of biomarkers in JSLE.

### 2.3. Data Extraction

Data related to the number of JSLE patients included in a particular biomarker study and the classification criteria used to diagnose JSLE organ involvement were extracted. Where there were statistically significant biomarker findings, the corresponding p value was incorporated in tables. Where there was a statistical correlation between biomarker levels and organ involvement, this was also included in tables, and the Pearson correlation coefficient r mentioned, with r = 0.3–0.5 indicating weak, r = 0.5–0.7 moderate and r > 0.7 strong correlations. Where available, area under the curve (AUC) for the receiver operating characteristics (ROC) of particular biomarkers was also included, with AUC values of 0.6–0.7 indicating poor, 0.7–0.8 moderate, 0.8–0.9 good and >0.9 excellent predictor value.

## 3. Autoantibodies as Biomarkers in JSLE

Autoantibodies to various self-antigens are the oldest and most studied biomarkers in SLE. The most sensitive and specific autoantibodies, such as anti-nuclear antibodies (ANA), anti-double-stranded deoxyribonucleic acid (dsDNA), anti-Smith (Sm), anti-cardiolipin and anti-beta 2 glycoprotein 1 antibodies, as well as lupus anticoagulant (LA) measured as prolonged Russell’s Viper Venom Time (RVVT) test were routinely measured in practice to facilitate SLE classification according to various criteria [23,24].

Although very few of these autoantibodies have been shown to have undeniable pathogenic properties in SLE, it is widely recognized that SLE is associated with impaired central or peripheral B-cell tolerance and enhanced circulating autoreactive B cells. Many of the SLE auto-antigens are immunostimulatory, leading to activation of toll-like receptor (TLR)7 or TLR9 and subsequently promoting plasma cell differentiation and auto-antibody production [25]. 

The biomarker value of autoantibodies in adult-onset SLE has been studied extensively as showed to be influenced by various factors, such as sensitivity, specificity and tendency to fluctuate, as well as the sensitivity of various assays: e.g. ANA are associated with increased sensitivity, but low specificity, while anti Sm and anti-dsDNA antibodies have increased specificity and low sensitivity for SLE, while anti-dsDNA antibodies tend to fluctuate with no correlation with clinical activity [26,27]. There is evidence that some autoantibodies, such as ANA and anti-dsDNA antibodies can predate SLE onset with many years [28,29], and that overall patients with JSLE had higher titres of anti-dsDNA, anti Sm and anti-ribonucleoprotein (RNP) antibodies that patients with adult-onset SLE as well as more severe disease [12].

Children with JSLE present with ANA positivity in 95%, but this test has a low specificity for SLE diagnosis, which can be as low as 36% [30], as ANA can be found in 15% of healthy children [31]. 

Research into the role of autoantibodies as JSLE biomarkers showed that although autoantibodies such as anti-complement fraction 1q (anti C1q), anti-dsDNA and anti-chromatin/nucleosome antibodies were significantly higher in patients with JSLE compared to healthy controls, and in JSLE with any level of disease activity compared to remission, they did not associate with renal involvement [32]. 

Older studies in children with JSLE identified association of anti-ds-DNA antibodies with active disease, arthritis, and rash, but not with renal involvement [33], while anti-Ro and La antibodies positivity correlated with cardiac involvement [34] and anti-cardiolipin antibodies fluctuated with various manifestations and were higher in children with neurological involvement [35].

In addition, there is evidence for ethnic differences in the prevalence of various JSLE associated autoantibodies. A study found that anti-U1RNP and anti-Sm antibodies were more prevalent in non-Caucasian JSLE patients [*p* < 0.0001 and *p* < 0.01, respectively [36]]. This cluster analysis found that JSLE patients with anti-dsDNA, antichromatin, anti-ribosomal P, anti-U1RNP, anti-Sm, anti-Ro and anti-La autoantibody were characterised by a higher prevalence of nephritis, renal failure, serositis, and hemolytic anaemia. This suggests a potential role of autoantibodies in defining a certain JSLE clinical phenotype rather than organ-involvement. 

Cluster analysis of autoantibodies in adult onset SLE also showed their association with certain clinical phenotypes and differential risk of damage, with SLE patients with anti-dsDNA, LA and anti-cardiolipin antibodies having the higher prevalence of cerebrovascular damage [37]. 

Despite the extensive clinical use of autoantibody measurements in diagnosing and classifying patients with SLE across age groups, future research is required to assess their role in personalised medicine or as predictors for response to a certain therapy [38]. Limited evidence for the role of high anti dsDNA antibody titres and low complement 3 (C3) as predictor biomarkers for response to therapy is available [39]. 

We describe below the main biomarkers identified in the literature, classified based on JSLE organ involvement.

## 4. Renal Biomarkers

The need for adequate biomarkers to help diagnosing and monitoring lupus nephritis (LN) is significant, given the prevalence and severity of renal involvement in JSLE. The current gold standard for diagnosis and classification of LN is the renal biopsy, which presents some challenges. A renal biopsy is an invasive procedure with potential risks of complications, such as bleeding and infection. In the paediatric population, biopsies are often performed under general anaesthesia, which could be associated with additional complications and logistical challenges. Furthermore, the quality of the renal tissue sample obtained through a needle biopsy has significant impact on LN diagnosis and classification [40], which is also influenced by inter-observer variability and dependent on the histopathologist expertise [41]. Whilst a renal biopsy can indicate the current level of disease activity, it is quite impractical and challenging to organise repeated biopsies, and therefore the renal biopsy has limited value in assessing LN dynamic, including worsening disease, response or resistance to treatment or the degree of accumulating chronic renal damage. Although renal biopsy is recommended for diagnosis of LN, the long-term monitoring of renal disease relies upon non-invasive assessments and scores, including the renal domains of well-validated disease activity scoring systems used in both JSLE and adult-onset SLE, such as the British Isles Lupus Assessment Group (BILAG) which assesses blood pressure, renal function and urinary protein, blood and sediment [42] and the Systemic Lupus Erythematosus Disease Activity Index (SLEDAI) which assesses urinary casts, haematuria, proteinuria and pyuria [43]. However, these scoring systems rely on non-specific features of renal dysfunction, do not differentiate between active LN and chronic renal dysfunction and are not able to identify patients at higher risk of disease progression, relapse or treatment-resistance. 

As such, there is a need for more versatile LN biomarkers, which ideally should not be affected by declining renal function, including low eGFR and proteinuria, or by the use of potentially nephrotoxic medication.

We summarise below the most commonly investigated (Table 1), as well as validated and non-validated novel biomarkers for LN (Table 2 and Table 3), appraising the level of evidence regarding their involvement in the disease pathogenesis and clinical relevance. 

### 4.1. Classical Renal Biomarkers

Complement activation appears to be central to the pathophysiology of SLE and, due to their cleavage to active C3b and C4b, decreased levels of C3 and C4 are seen in SLE flares and are also included in the classification criteria for SLE [10,23,44]. The evidence for supporting the role of serum C3 and C4 measurements as biomarkers of LN in JSLE is mixed; three studies found no significant differences in C3/C4 levels in active LN whilst four studies demonstrated significantly lower levels of C3/C4 in active LN [44,45,46,47,48,50]. This variability could be due, in part, to the use of different classifications for definition of a renal flare: e.g. studies that used BILAG renal domains to define active LN tended to find lower C3/C4 in active disease, whereas studies that used SLEDAI renal domain or the biopsy score to assess for presence of active LN did not.

Antibodies against double-stranded DNA [dsDNA] are a highly specific feature of JSLE and are also included in the classification criteria [23]. Interestingly, although historically considered a reliable marker for renal flares in SLE, four studies in JSLE population found no difference in dsDNA levels in active LN compared to other disease manifestations, whilst only one study demonstrated significantly elevated dsDNA levels in children with LN with a poor predictor value for active LN [44,45,46,48,50]. The role of dsDNA as a biomarker for LN flares is likely to be more nuanced, as data from adult SLE studies suggest significant variability of dsDNA titres around the time of renal flare; i.e., an initial increase in dsDNA before the flare, followed by a decrease during the flare [52].

There is evidence that levels of C-reactive protein [CRP] usually remain stable during LN flares in JSLE [46]. Conversely, erythrocyte sedimentation rate [ESR], a non-specific test reflecting both active inflammation and disease pathophysiology [potentially related to polyclonal hypergammaglobulinemia, increase in fibrinogen irrespective of disease activity, as well as by anaemia associated with chronic inflammation or haemolysis], was significantly elevated in flares of LN and was a fair predictor of renal flares in JSLE [44,45,46].

Proteinuria is the hallmark of renal disease due to glomerular, or less commonly tubular, dysfunction and the presence of proteinuria, even in patients with a normal eGFR, is associated with adverse outcomes, including progression of renal disease, need for renal-replacement therapy, myocardial infarction and death [53]. Proteinuria is included in the renal BILAG score [42] and measured either via urine dipstick, urinary albumin/creatinine ratio [ACR], urinary protein/creatinine ratio [PCR] or 24-hour urine protein. On the other hand, in the SLEDAI score it is defined as urinary protein greater than 0.5 g in 24 h [43]. It is not surprising that multiple studies have shown increased proteinuria during active LN in JSLE using both the BILAG and SLEDAI criteria [44,45,46,49,50]. Studies that used renal biopsies to diagnose active LN have shown variable results, potentially influenced by the heterogeneity of biopsy scoring systems used in various JSLE studies [47,50,51]. Although a useful sign of renal dysfunction and adverse outcome, proteinuria is not a specific feature of LN and could persist after a renal flare has subsided in 61% of JSLE patients [51]. As such, the use of proteinuria in isolation as biomarker to guide management in LN appears to have poor performance, especially when compared to other urinary biomarkers, as detailed below. However, proteinuria still has significant utility in clinical practice, especially in combination with other markers, as it is sensitive, easy and quick to detect, cheap and readily available in clinical settings across the world and can be repeated as clinically indicated.

Serum creatinine and eGFR are included in the renal domain of the BILAG score but not in the SLEDAI score. However, multiple studies have demonstrated no significant difference in creatinine or eGFR in patients with active LN [especially in patients with no background renal damage] and both measurements were not useful, as expected in predicting JSLE disease activity in patients with good renal reserve [44,45,46,47,48,49,50]. Creatinine can vary significantly with muscle mass and whilst the eGFR attempts to adjust for this, both creatinine and eGFR are useful for assessing renal dysfunction due to chronic damage by LN, but they are poor biomarkers of active disease (Table 1).

### 4.2. Validated Novel Urinary Renal Biomarkers

#### 4.2.1. NGAL

Neutrophil gelatinase-associated lipocalin (NGAL) is a protein that binds to iron-chelating molecules and is found in renal epithelial cells [61]. It is produced in response to acute renal injury and is thought to have a reno-protective role through iron chelation and transport mechanisms. Increased expression of NGAL by renal epithelial cells in response to renal injury resulted in a 6-fold increase in urinary NGAL (uNGAL) concentration, which started 2 days before acute kidney injury [AKI] onset in critically ill pediatric patients and was predictive of severity of renal injury [62]. Meta-analyses have found NGAL to be an early predictor of AKI, which can be used to detect AKI in many settings [including after cardiac surgery, post-contrast and in critically unwell patients], can predict initiation of renal replacement therapy (RRT) and mortality with a higher predictive value in children than in adults [63]. The role of NGAL as a biomarker for JSLE nephritis was first studied in 2006; uNGAL was shown to be higher in patients with JSLE than in positive controls with juvenile idiopathic arthritis (JIA), another chronic inflammatory rheumatic condition diagnosed in children, which is not associated with renal involvement [56]. The same study demonstrated that uNGAL levels were 16-fold higher in patients with biopsy-proven LN than JSLE patients without renal involvement. As well as being a marker of LN, uNGAL has demonstrated its potential application as a marker of LN activity; uNGAL levels (both absolute concentrations and urine-creatinine adjusted concentrations) have been shown to be elevated in active LN and correlated weakly with LN activity [44,50,56,57,58,64]. Increased levels of uNGAL are seen with worsening LN and can predate a LN flare by 3–6 months [55,64]. Urinary NGAL levels demonstrated an excellent ability to discriminate between patients with and without biopsy-proven LN [56], was able to predict proliferative LN and was a good predictor of worsening SLE renal activity measured clinically [55]. Urinary NGAL measurements were reliable as they did not vary with patients’ height, weight, age, sex or race [2,55], did not correlate to creatinine clearance or proteinuria [58] and were not affected by extrarenal disease activity [58,65] suggesting that they were quite specific for renal involvement in JSLE. Despite the good biomarker value of uNGAL, there were no significant changes in serum NGAL levels during flares of LN [58,65], although a rise in plasma NGAL levels [44] was predictive of a worsening global disease activity score [55] (Table 2).

#### 4.2.2. MCP-1

Monocyte chemo-attractant protein-1 (MCP-1) is a chemokine produced by a large variety of cells that mediates the recruitment of monocytes from the bone marrow into the blood stream, drives the migration of monocytes through a chemokine gradient into areas of active inflammation and induces differentiation of monocytes locally, leading to further production of inflammatory cytokines [66]. MCP-1 is produced by tubular epithelial cells resulting in increased cytokine and adhesion molecule expression which further activates tubular epithelial cells, increasing the inflammatory response [65]. Animal models of SLE have demonstrated the role of MCP-1 as a driver of inflammatory cell infiltration and have suggested a potential therapeutic target role, as MCP-1 deletion reduced renal inflammation and proteinuria, and prolonged survival [67,68]. As a urinary biomarker, MCP-1 level has been demonstrated to increase in JSLE LN flares, to be a fair-good predictor of LN activity and to be higher in non-responders to therapy [44,45,48,50,54]. MCP-1 levels have also been shown to be higher in JSLE patients with renal flares compared to SLE patients with non-renal flares, healthy controls or renal disease controls. Moreover, higher levels were noted in patients with severe renal flares (vs moderate or mild), impaired renal function during flares, and proliferative LN [class III/IV] compared to non-proliferative (class V) [69] in SLE with adult-onset. As such, MCP-1 has a role not only in the detection of LN, but also may reflect the severity of flare and response to treatment (Table 2). Future research is required to assess its role as a therapeutic target in LN in children with JSLE.

#### 4.2.3. AGP-1

Alpha-1-acid glycoprotein (AGP-1) is an acute phase protein primarily produced by hepatocytes in response to inflammation and infection driven by cytokine production, primarily interleukin (IL)-1, IL-6 and tumour necrosis alpha (TNFα), and glucocorticoids, including exogenous dexamethasone [70,71]. Additionally, AGP-1 appears to be produced by epithelial cells and leukocytes within various organs including the kidney [72,73,74]. AGP-1 appears to mediate natural homeostasis by providing negative feedback to reduce inflammation [75]. Animal models have demonstrated that AGP-1 can reverse renal fibrosis and inflammation and preserve tubular epithelial structure, and as such it has been suggested to have a potential therapeutic role in LN [76,77]. As a urinary biomarker, AGP-1 levels were higher in patients with JSLE compared to JIA controls, and in particular in JSLE patients with LN flares [11,46,48,49,60], correlated to LN activity and could predict with fair-good accuracy a LN flare [47,55,59]. AGP-1 levels were higher when crescents, tubular cell necrosis and mesangial proliferation were present on renal biopsy [47,48]. Furthermore urinary AGP-1 levels increased at least 3 months before a clinically detectable flare of LN and could predict a flare up to 12 months in advance [49,59]. AGP-1 is also an excellent predictor of non-responders to therapy [54]. As such, monitoring urinary AGP-1 levels in JSLE may allow for early detection of LN and severity stratification and assessment of treatment response. Serum levels of AGP-1, however, were not different from JIA controls and did not differ in patients with active LN versus patients with no renal involvement [49].

#### 4.2.4. Ceruloplasmin

Ceruloplasmin (CP) is a copper containing feroxidase which is synthesised by the liver and acts as an antioxidant (by oxidising Fe^2+^ to Fe^3+^ and reducing the oxidative damage to tissues). Additionally, CP is an acute phase protein, rising rapidly in inflammation [78]. It has been found to have a role in Wilson’s disease, as well as neurodegenerative, demyelinating and cardiovascular disease [79]. Animal models have demonstrated a renal protective role of CP as its absence resulted in iron deposition in the renal medulla and cortex, structural renal abnormalities and subsequent proteinuria [80]. Urinary levels of CP are higher in JSLE than in JIA patients, and increased during LN flares, although not when adjusted for urinary creatinine or protein levels [44,48,49,60]. CP is a poor-good predictor of a LN activity but a good predictor of LN damage [49]. Additionally CP was a good predictor of lack of response to therapy, and its decrease could predict remission 12 months before this was clinically detectable [54,59]. Serum CP were similar in JSLE and JIA controls and did not rise in active LN flares [49]. As such, CP has potential as a urinary biomarker of LN and may be useful in predicting treatment response, allowing for immunosuppressing therapies to be adequately optimised. Additionally, due to its role in the screening for Wilson’s disease, validated ELISA kits for CP measurement are widely available in hospitals.

#### 4.2.5. L-PGDS

Lipocalin-type Prostaglandin-D Synthetase (L-PGDS) is an enzyme which is secreted by the brain, heart and male genital organs into the cerebrospinal fluid, serum and semen respectively [81]. L-PGDS has two functions; it acts as an enzyme to produce prostaglandin-D2 (PGD2) and it binds to lipophilic proteins in extracellular spaces such as retinoids, thyroid hormones, and bile pigments. L-PGDS is the second most common protein in the cerebrospinal fluid where it has a role in sleep regulation, nociception, and potentially in the pathogenesis of neurological diseases [82,83]. L-PGDS is produced by the endo- and myocardium in patients with coronary artery disease and increased levels have been detected in the urine of patients with hypertension, especially in those with renal dysfunction [84,85]. Less is known about the role of L-PGDS in renal JSLE, although animal models have demonstrated that urinary L-PGDS levels increased in acute renal injury and preceded albuminuria [86], whilst human studies have found detectable increases in urinary L-PGDS in diabetic nephropathy with potentially increased sensitivity compared to proteinuria [85]. In the kidney, L-PGDS increases renal artery flow, renin activity and renal function [87]. Urinary levels of L-PGDS were higher in JSLE patients compared to JIA controls [49] and were higher in active vs. inactive, when the renal domain of BILAG score was used for appreciating the disease activity rather than the renal biopsy [44,48,49,60]. L-PGDS was weakly correlated with LN activity, had a good-fair predictive value for LN activity and acted as a good predictor for LN damage in JSLE [47,60]. Furthermore, L-PGDS levels increased at least 3 months before a LN flare and were an excellent predictor of non-responders to LN therapy [49,54]. Notably, while creatinine-adjusted urinary L-PGDS increased in active LN, protein-adjusted urinary L-PGDS levels fell. However, serum L-PGDS level was not increased in JSLE patients compared to JIA controls or during LN flares [49]. L-PGDS is a promising urinary biomarker of active LN, especially for its ability to predict renal disease activity and identify potential non-responders to therapy, although its lack of specificity for JSLE may hinder its clinical applications in patients with other comorbidities, such as diabetes or hypertension.

#### 4.2.6. Transferrin

Transferrin is an abundant iron-binding glycoprotein produced by the liver involved in iron transfer, storage and delivery to tissues. Like albumin, transferrin is a small, negatively charged protein with similar glomerular excretion rates and kinetics of excretion to that of albumin [88]. After glomerular filtration, transferrin reuptake occurs in the proximal tubule and collecting ducts through transferrin receptors, to facilitate iron transport necessary for cell metabolism [89]. Urinary transferrin has been demonstrated to predict microalbuminuria in diabetes, an early sign of diabetic nephropathy [90]. Urinary transferrin was significantly higher in JSLE patients compared to JIA controls [48]. Studies have demonstrated that urinary transferrin levels were higher during LN flares [49,60], although one study only demonstrated statistical significance in one JSLE cohort (US) but not in another (UK) [44]. Another study found increased levels in LN diagnosed on renal biopsy by using the National Institutes of Health-Activity Index NIH-AI [91], a commonly used LN classification that focuses on glomerular pathology, rather than the Tubulointerstitial Activity Index (TIAI) [92], which focuses on tubulointerstitial pathology [48]. The increase in urinary transferrin in LN flares was also detected when adjusted for urine creatinine or protein levels, suggesting that it reflects more than just increased glomerular permeability [49]. Overall, urinary transferrin was a fair-good predictor of LN activity and chronicity in JSLE [47,49]. In contrast, serum transferrin levels were similar in JSLE patients versus JIA controls or in JSLE patients with our without LN flares [49]. Urinary transferrin levels increased at least 3 months before an active JSLE LN flare could be detected clinically [49]. Furthermore, urinary transferrin was an excellent predictor of patients who do not respond to therapy. As such, despite some conflicting evidence, urinary transferrin appears to have clinical benefit as an early biomarker for LN, by being able to identify individuals who may need more intense treatment to achieve remission. Further research is needed to establish the pathophysiology of transferrin leakage into urine in LN and validate this biomarker in other JSLE cohorts.

### 4.3. Other Novel Renal Biomarkers Investigated

There are numerous other renal biomarkers that I have investigated in renal JSLE, although not validated (Table 3).

Vascular Cell Adhesion Molecule-1 (VCAM-1) is a cell surface protein that leads to the adherence of leukocytes to the vascular endothelium to mediate inflammatory cell infiltration and is expressed in the cortical tubules and glomeruli in animal models of LN [97]. In adults, VCAM-1 was elevated in active LN but also in other types of glomerulonephritis [including ANCA-associated glomerulonephritis, membranous glomerulonephritis and focal segmental glomerulosclerosis] [98]. There is limited data in JSLE, although VCAM-1 has been demonstrated to be elevated during LN flares [44,54]. Further research is needed to test its ability to predict LN and to establish its clinical value.

Adiponectin is a protein produced by adipose tissue that regulates glucose control and is implicated in the development of type 2 diabetes and cardiovascular disease in obese patients [99]. It also has an anti-inflammatory role and is expressed on podocytes, tubular cells and tubular casts in active LN [100]. Urinary adiponectin was elevated in LN flares and acted as an excellent predictor of response to therapy [48,54]. 

Hemopexin is a protein produced by the liver that binds heme to protect the body from oxidative damage, which accumulates in the cortex in renal injury [101]. Urinary levels of hemopexin were significantly higher during JSLE LN flares and hemopexin was a good predictor of non-responders to LN therapy [48,54]. 

Hepcidin is another protein involved in iron metabolism that reduces iron availability in inflammation and is central in the pathogenesis of anaemia of chronic disease [102]. Urinary levels of hepcidin have not been demonstrated to increase in active LN compared to extra-renal JSLE, although levels were higher in non-responders to therapy, possibly reflective of a higher inflammatory state in these individuals [48,54].

Kidney Injury Molecule-1 (KIM-1) is a protein that is released by renal tubular cells in response to injury to promote phagocytosis of apoptotic cells and cellular debris and drive the repair of injured renal tubules, and as such is a urinary biomarker of AKI and chronic kidney disease (CKD) [103]. Its level was increased in active LN and is a good predictor of non-responders to therapy for LN in JSLE [48,54].

Transforming Growth Factor-Beta [TGF-β] is a cytokine that has been described as the ‘master regulator’ of renal inflammation and fibrosis, and is primarily responsible for driving the fibrosis associated with CKD [104]. Higher levels of TGF-β have been found in the urine of JSLE patients with active LN and urinary TGF-β is a good predictor of non-response to therapy [48,54]. Further research is required to investigate the ability of urinary TGF-β to predict chronic renal damage and into therapeutics that target this protein. 

Osteopontin is another cytokine that is upregulated in inflammation and correlates with glomerular disease and albuminuria in diabetic nephropathy [105]. However there was no difference detected in urinary osteopontin levels between JSLE patients with and without renal involvement [54]. 

Interferon gamma-induced Protein 10 (IP-10) is another pro-inflammatory cytokine which was investigated in JSLE nephritis, although there were no differences between active compared to non-active LN JSLE patients or between JSLE patients and healthy controls [50].

Vitamin D Binding Protein (VDBP) is a vitamin D transporter protein. Vitamin D is freely filtered through the glomeruli and reabsorbed by proximal tubular cells where it is hydroxylated to the active form of vitamin D [106]. Urinary levels of VDBP were higher in active LN, likely because tubular injury is associated with reduced reabsorption of VDBP [48]. Furthermore, VDBP was an excellent predictor of response to JSLE LN therapy [54].

Liver-type Fatty Acid Binding Protein (L-FABP) is a reno-protective protein that promotes the excretion of lipid peroxidation products. It is an emerging urine biomarker in renal disease, as elevated urinary levels were detected in AKI, contrast nephropathy, glomerulonephritis and diabetic nephropathy [107]. Its role in JSLE is less clear; one study found no difference in active versus inactive LN in JSLE, while another found it to be a good predictor for inadequate response to therapy [48,54].

The S100 protein family is a large group of calcium binding proteins with a broad range of functions; it is thought that dysregulation and overproduction of S100 proteins may be involved in the pathogenesis of autoinflammatory conditions and S100 proteins have been suggested as potential biomarkers in many autoinflammatory conditions [108]. Studies in JSLE have shown mixed results; one study found that serum S100A8/A9 and S100A12, and urinary S100A12 were significantly higher in JSLE patients (compared to JIA controls) and in patients with active versus inactive renal disease [93], while another study showed that although urinary S100A4, S100A6, S100A8/A9 and S100A12 were all elevated in active LN and fell with resolution of LN, the serum levels of S100 proteins did not correlate with disease activity [94]. Urinary S100A4 was higher in proliferative LN (class III/IV) compared to membranous LN (class V). Moreover, immunohistochemistry demonstrated S100A4 staining in mononuclear cells, podocytes and distal tubular cells in those with LN and controls, which further supports its potential role in the kidney.

Cystatin C is a small protein that is produced by all nucleated cells and freely filtered in the kidney. It is less influenced by age, muscle mass or diet compared to creatinine. There has been a large amount of research into the role of cystatin C as a biomarker of renal function and it has been shown to outperform creatinine in estimating eGFR in AKI and CKD, especially in early stages of CKD [109]. Unfortunately, it has yet to demonstrate promise in JSLE. Urine cystatin C levels did not increase in LN flares although urine cystatin C was a fair predictor of non-response to therapy and serum cystatin C weakly correlated with LN damage [48,54,94]. Although potentially promising, further research is required to demonstrate the role of cystatin C as a biomarker in early chronic renal damage in JSLE. 

The soluble alpha chain of the interleukin-2 receptor (also called soluble CD25—sCD25) is produced by the proteolytic cleavage of interleukin 2 receptor (IL-2R) subunit alpha from cell membrane, following T cell activation. sCD25 has been found to be a suitable serum and urine biomarker for LN in adult-onset SLE [110,111], and more recently in JSLE, correlating positively with the level of proteinuria, and the SLEDAI, renal SLEDAI, and SLICC renal activity scores, and negatively with C3 serum levels [95]. In this study, the normalised urinary sCD25 levels were higher in JSLE patients than in healthy controls, and in JSLE patients with active LN compared to active JSLE patients without LN [95].

Natural Killer (NK) cells are components of the innate immune response which and can be subdivided into CD56^dim^ NK cells, which accounts for 90% of circulating NK cells and have a central role in cell cytotoxicity, and CD56^bright^ NK cells, which make up the remaining 10% and have an immunomodulatory role [112]. Peripheral blood levels of CD56^bright^ NK cells strongly correlated with LN activity and moderately correlated with LN chronicity, although urinary levels have not been studies and flow cytometry is needed for quantification [96].

#### 4.3.1. Renal Biomarker Combination Panels

Whilst individual urinary biomarkers have been demonstrated to correlate with LN activity, no single biomarker has demonstrated an excellent ability (AUC > 0.9) to predict active LN. MCP-1, NGAL, AGP-1, ceruloplasmin and transferrin were good predictors (AUC > 0.8), L-PGDS and ESR were fair predictors (AUC > 0.7) and C3/C4 and dsDNA were poor predictors (AUC > 0.6) of LN activity (Table 1, Table 2, Table 3 and Table 4). As such, numerous studies have tested combinations of various biomarkers (Table 4). Many of these combination panels have demonstrated an excellent ability to discriminate between active LN and absence of LN, with one panel including AGP-1, ceruloplasmin, L-PGDS and transferrin demonstrating a perfect ability (AUC 1) to predict LN activity, reaching 100% sensitivity and specificity(44). As such, urinary biomarker panels are an excellent way of non-invasively monitoring LN activity and, should they become commercially available, would improve personalised care for patients with JSLE nephritis.

Overall, the afore-mentioned renal biomarkers in JSLE could have potential clinical utility in the identification of LN, activity and damage assessment, as well as prediction of future LN flares and treatment response (Figure 1).

Future research should be focused on identifying response biomarkers to a certain therapy, to enable JSLE patient selection for inclusion in clinical trials. Recent research in adult-onset SLE showed that a urinary panel including LPGDS, transferrin, AGP-1, ceruloplasmin, MCP-1 and sVCAM-1 predicted response to rituximab treatment at 12 months (AUC 0.818) [113].

Although research into biomarkers predicting other type of organ-involvement in JSLE is less advanced that in JSLE nephritis, we present below examples of validated and non-validated biomarkers for prediction of other JSLE manifestations (Table 5).

#### 4.3.2. Cardiovascular Biomarkers

Patients with adult-onset SLE are at an increased risk of developing cardiovascular disease (CVD) through rapidly progressive atherosclerosis compared to healthy individuals of the same age and gender [123,124]. JSLE has an increased CVD and mortality risk compared to adult-onset SLE [12,125,126]. Investigating the development of atherosclerosis and CVD in these younger, more severely affected patients, has potential clinical value in identifying early mechanisms of vascular damage and biomarkers to guide management strategies [114]. Whilst traditional risk factors such as hypertension, diabetes and high cholesterol contribute to the increased cardiovascular risk (CVR) in SLE overall, they fail to fully account for it [127]. Ideally, all patients with SLE should receive monitoring and treatment for modifiable CVR factors [114,126]. The interplay between traditional CVR factors, increased disease associated inflammation and treatment burden in these younger patients could contribute to early, accelerated atherosclerosis in JSLE [125,127,128]. Despite this, no guidelines exist for CVR management in JSLE and it is not possible to predict the CVR of JSLE patients using traditional risk factors.

The most established and direct methods of quantifying CVR is through non-invasive vascular measurements. These include ultrasound methods such carotid intima media thickness (CIMT) and flow mediated dilation (FMD) and electrocardiogram methods such as pulse wave velocity (PWV), which can detect subclinical atherosclerosis and assess endothelial function. Assessing the arterial functional impairment led the way to identify early evidence of CVD in JSLE. Various serological and imaging biomarkers have been tested in a quest to establish the best ones likely to associated with clinical relevance for increased CVR in JSLE patients (Table 5). In a study of 45 children with JSLE a significantly higher CIMT was detected compared to matched healthy controls (HCs) [0.45 vs. 0.37 mm, respectively, *p* < 0.0001] [114]; however the PWV was not shown to be significantly different in JSLE compared to HCs. These findings have been validated in a separate study of 24 JSLE patients which showed increased CIMT and PWV in JSLE patients compared to age and sex-matched HCs [115]. Increased PWV in JSLE patients was also identified in a larger study including 88 patients [129]. Because of the large heterogeneity of the disease presentation, duration and impact of medications used, not all studies validated these findings [130,131]. Although less used in JSLE, cardiac MRI has been associated with good sensitivity for the detection of silent CVD with structural and functional myocardial impairment in patients with adult-onset SLE even at diagnosis [132] or in the absence of clinical criteria for cardiac involvement [133]. Longitudinal studies are needed to provide answers related to clinical implications of monitoring subclinical atherosclerosis and managing CVR in these patients from an early age.

Studies have demonstrated that changes in CIMT and PWV can predict future cardiovascular events [134]. Despite the variability of findings in various JSLE cohorts, it became clear for clinicians and scientists that vascular assessments were useful tools to measure the development of subclinical atherosclerosis in JSLE and could be used as endpoints in studies that seek to validate new biomarkers. A study employing a multivariate analysis showed that lymphopenia was consistently associated with the progression of atherosclerosis assessed by CIMT in JSLE [116]. Another study showed that nephrotic range proteinuria was associated with significantly increased CIMT, however, this association was probably confounded by the fact that these JSLE patents also had increased SLEDAI score and higher levels of serum cholesterol [135]. 

Clinical trials targeting atherosclerosis progression by use of statins have reported mixed results in SLE. Some have shown a reduction in vascular inflammation and mortality in adult patients [136,137], whilst others showed no effect on the progression of subclinical atherosclerosis in adult-onset SLE [138], as well as JSLE [139]. The success of future trials will likely depend on correct stratification of patients and therefore it is important to invest in research leading to the discovery of more precise CVR biomarkers which can drive personalised therapeutic approaches.

Despite not meeting the primary endpoint, which was reduced CIMT in JSLE patients treated with atorvastatin versus placebo, the secondary analysis of the APPLE trial found evidence that statins may reduce the progression of atherosclerosis in JSLE patients with lower vitamin-D levels and higher levels of high sensitivity CRP [140,141]. Several studies have supported a role for CRP as a biomarker for CVR detection in JSLE [116,117]. This highlights the importance of combining conventional routine measured biomarkers for CVR detection as well as patient stratification strategies using new biomarkers in the design of interventional clinical trials addressing CVD in JSLE. Follow up studies to investigate the pathogenic relevance of vitamin D in atherosclerosis progression in JSLE are highly warranted.

Dyslipidaemia, a conventional risk factor for atherosclerosis and CVD, is a common feature of patients with both adult-SLE and JSLE, and includes elevated triglycerides and apolipoprotein(Apo)-B expressing low-density lipoproteins (LDL), alongside reduced high-density lipoproteins (HDL) expressing ApoA1 [142,143,144,145]. Evidence suggests that lipoprotein biomarkers could be used predict atherosclerotic risk in adults with SLE [146,147]. A study using in-depth metabolomics by Robinson et al. identified that the ApoB:ApoA1 ratio was a strong biomarker of CVD in JSLE, independent of clinical disease measures and body mass index (BMI) [118]. This finding has been validated in 2 separate cohorts using ultrasound evidence for the presence of subclinical atherosclerotic plaque in adult SLE patients [146]. A multivariate linear regression study investigating laboratory lipid biomarkers from JSLE patients recruited to the APPLE trial identified a positive association between SLEDAI, prednisolone dose, hypertension and renal injury (proteinuria and creatinine) and higher LDL levels, and a negative association between higher CRP and creatinine levels with HDL in JSLE patients [148]. Together, these findings suggest that serum biomarkers associated with dyslipidaemia could hold promise for patient stratification in JSLE to improve the success of therapies targeting lipid metabolism to reduce CVR in these patients. 

As atherosclerosis is a well-defined inflammatory process, there is an interest in research investigating inflammatory biomarkers of CVD in JSLE. Endothelial cells present within the lining of large blood vessels allow for vascular integrity, function, repair and protection against atherosclerosis. Endothelial progenitor cells have been shown to be reduced in number and function in JSLE, largely through impaired differentiation to mature endothelial cells, suggesting an increased CVR through vascular damage [149]. This could be due to an increased cellular production of inflammatory cytokines. Several studies in JSLE have found a significant positive correlation between TNF-α levels (a strong pro-inflammatory cytokine) and atherogenic biomarkers including increased triglyceride levels and expression of adhesion molecules on endothelial cell membranes (which are involved in recruitment of inflammatory cells to atherosclerotic plaques) [117,150]. Inflammatory cytokines produced by adipocytes (adipokines) can also alter lipid metabolism and atherosclerotic plaque development. A study in JSLE highlighted a positive correlation between adiponectin levels and circulating cholesterol, suggesting that adipokines could be novel biomarkers of CVR associated with JSLE [119]. 

Despite progress achieved in understanding the pathogenesis of CVD in JSLE and identification of biomarkers with potential use for patient stratification, future studies are required to establish the pathogenic link between various biomarkers and clinical endpoints, as well as investigate their usefulness in guiding personalised CVR management strategies in JSLE.

## 5. Central Nervous System [CNS] Biomarkers

Neuropsychiatric systemic lupus erythematous [NPSLE] is often considered a ‘diagnosis of exclusion’ [151,152], and it encompasses a wide range of both psychiatric and neurological syndromes observed in SLE patients [121,151]. Differentiating NPSLE from an alternative aetiology that is unrelated to SLE is difficult and poses a diagnostic challenge [151]. 

NPSLE is incredibly heterogenous, and thus clinical presentations are diverse, and can include headaches, mood disorders, cognitive dysfunction, psychosis, seizures, and cerebrovascular disease, with the American College of Rheumatology (1999) identifying 19 neuropsychiatric syndromes that may occur in SLE [40,153,154,155,156]. The severity and presentation of CNS disease fluctuates over time and the literature suggests that it often occurs independently of disease activity in other organs and systems [157]. The pathophysiology of NPSLE is not fully understood but the disease manifestations are considered a direct consequence of either an underlying ischaemic, thrombotic or inflammatory process [158]. 

There is a higher prevalence of CNS involvement in patients with JSLE compared to patients with adult-onset SLE [7,155,156,159,160,161], and this is often indicative of a greater severity in the disease course and an increased morbidity and mortality in younger patients [7,156,161,162,163,164,165]. Because of the challenges in recognising and classifying NPSLE, current research is very limited and there is no ‘gold standard’ approach for the diagnosis and management of NPSLE in children. A few serological markers have been investigated in NPSLE (Table 5).

Antiphospholipid antibodies (APLAs) are a well-established marker of hypercoagulability and increased risk of thrombosis and focal neurological symptoms amongst adult patients with SLE, with the triple positive patients (positive for anticardiolipin and β2-glycoprotein 1 antibodies, as well as lupus anticoagulant-LAC) having the highest risk [157,158,166,167]. 

There appears to be a high prevalence of APLAs in children with NPSLE [7,167,168,169] with evidence to suggest a significant association with focal neurological symptoms (including cerebrovascular disease) [151,168], but the specific APLAs associated with this heightened risk remains unclear [1,151,158]. LAC [9,169] and anti-cardiolipin [1,162,167] antibodies are frequently mentioned in the literature as risk factors for vascular thrombosis in children with JSLE. However, data are lacking to suggest an association of APLAs with other frequently observed neurological manifestations of JSLE [151,167,168]. 

Anti-ribosomal P and anti-ganglioside M1 are other autoantibodies that have been reported in children with NPSLE, however the literature pertaining to their potential pathogenic role is conflicting. A recent prospective study by Hanly et al. found a significant association between anti-ribosomal P seropositivity and psychosis in adults with SLE [153]. Although the literature suggests that these autoantibodies are more prevalent in JSLE [7,151], no significant association with psychosis was found in paediatric studies to date [170]. Antibodies associated with autoimmune encephalitis, e.g., NMDAR antibodies measured by cell based assay, have not been found [171].

There is evidence that a small number of JSLE patients with antibodies against aquaporin 4 (a water channel) experienced neurological manifestations, and that these autoantibodies were correlated with increased likelihood to experience neurological symptoms (*p* = 0.002), less likelihood to have skin involvement (*p* = 0.045), or detectable anti-dsDNA antibodies (*p* = 0.022) [122]. Aquaporin 4 antibodies have been found in those with lupus and optic neuropathy and myelitis [122] and represent a treatable condition, neuromyelitis optica, and so are clinically useful investigations, and also antibodies to myelin oligodendrocyte glycoprotein (MOG) [120].

Cognitive impairment is a common clinical manifestation of NPSLE [159], and psychometric testing is a useful non-invasive method to assess and monitor cognitive dysfunction in patients with SLE [1,151]. However, this is a comprehensive and laborious set of tests that requires the input of a specialist neuropsychologist as it must be tailored to patients’ age. The Montreal Cognitive Assessment (MOCA) and the Mini Mental State Examination (MMSE) are cognitive screening tools that are used in the assessment of cognitive function in adults with SLE [151]. However, the latter is known to be an insensitive test in detecting mild cognitive dysfunction, and both tests have not been validated in children with JSLE. The Paediatric Automated Neuropsychological Assessment Metrics (PedANAM) has been identified as a useful screening tool that can be used by non-specialists to identify cognitive dysfunction in children with JSLE [9,172,173]. A reported limitation is that this test is lengthily and requires special computer software that is not always easily accessible [174]. 

MRI is the most utilised imaging modality in the investigation and diagnosis of juvenile NSPLE, and it is important in helping to rule out alternative diagnoses [151]. However, there is no single imaging finding that is diagnostic for NPSLE, and also MRI examinations can be normal, especially in patients with diffuse syndromes that include headaches, cognitive dysfunction, and psychiatric disease [166,175]. When neuroimaging abnormalities are present, common findings in JSLE patients include, white matter hyperintensities seen on T2-weighted images, cerebral atrophy, and small cortical infarcts [151,162]. A retrospective study by Yu et al. of 185 JSLE patients described abnormal MRI findings in 92.5% patients who developed NPSLE, with brain atrophy and infarction being the most common observed [162]. However, the prevalence of abnormalities has been reported as lower in other studies [175,176], and abnormal findings have also been observed in children prior to the onset of NPSLE, as well as in children with JSLE without CNS involvement. This suggests that MRI findings are generally non-specific in JSLE, while in some cases they could reflect sub-clinical or evolving NPSLE, requiring sequential imaging for diagnosis. 

Functional MRI (fMRI) is a type of ‘advanced brain MRI modality’ that is known to be more sensitive in detecting early microstructural brain damage. This technique is commonly used to measure brain functionality during cognitive and sensory testing; however, its use in NPSLE is limited and not yet validated. A small pilot study by DiFrancesco et al. (2007) observed differences in neuronal network activation patterns using fMRI in children with JSLE compared to healthy controls, highlighting it as a promising approach to further our understanding of the brain areas and mechanisms involved in the development and progression of cognitive dysfunction in JSLE [1,177]. A further cross-sectional study by DiFrancesco et al. (2013) identified an association between differences in brain activation patterns and specific changes in cognitive function in JSLE. Authors postulated that patients with JSLE initially compensate well for disease induced changes in brain function by increasing the activation of certain areas of the brain. However, cognitive dysfunction becomes clinically apparent when these compensatory mechanisms fail. These studies are suggestive that fMRI could potentially be used as a biomarker to predict and monitor cognitive dysfunction in this cohort of patients [178,179]. 

The pathogenesis of NPSLE is poorly understood but the heterogenicity of the disease suggests that the aetiology is diverse and multifactorial. The identification and correlation of biomarkers with specific phenotypes requires further research into understanding the pathophysiology and disease progression of NPSLE. This will ultimately help facilitate early diagnosis, targeted treatments, and improved monitoring of children with NPSLE.

## 6. Arthritis Biomarkers

The musculoskeletal (MSK) system is commonly affected in patients with SLE [180,181,182,183], and involvement can vary from mild arthralgia to a deforming arthropathy in rare cases [180,183,184]. However, arthritis in children with JSLE is most commonly polyarticular and almost always non-deforming and non-erosive [180]. 

A variable proportion between 20–30% patients with adult-onset SLE are positive for rheumatoid factor and this correlated with clinical features of arthritis [185,186]. In contrast, a smaller proportion of adults with SLE have been found positive for anti-citrullinated cyclic peptides (CCP) antibodies, but this biomarker was associated with erosive arthritis in adults with SLE [187], in particular in SLE patients with higher titres [188]. Because of lower prevalence of arthritis in JSLE compared to adult-onset SLE [12], there are no studies looking into the diagnostic and prognostic value or RF and anti CCP antibodies in JSLE. 

Recent studies have investigated the potential role of musculoskeletal ultrasound (MSUS) in the evaluation and monitoring of articular [183,184,189,190] and periarticular [184,191] changes in adults with SLE, both with or without clinical evidence of MSK involvement. A study by Yoon et al., demonstrated subclinical synovitis in 58.3% (28/48) of SLE patients, with a significant positive correlation between ultrasound severity index scores (USSI) and ESR levels (r = 0.30, *p* < 0.05) and anti-dsDNA antibody titres (r = 0.34, *p* < 0.05). Higher USSI scores (OR 12.93, 95% CI 1.023–163.503, *p* = 0.048) were also independently associated with the development of new MSK symptoms in these individuals [189]. There is also evidence that MSUS has a role in guiding therapeutic decisions, as SLE patients with MSUS detected subclinical synovitis respond better to steroid treatment [192]. 

There are very few studies evaluating the use of MSUS in children with JSLE. One study by Demirkaya et al., identified decreased flexor tendon thickness at metacarpophalangeal joint level in children with JSLE compared to healthy controls, however tendon thickness did not correlate with JSLE disease duration or activity [*p* > 0.05] [193]. Further research is needed to verify and standardise the role of US in the assessment and management of both adults and children with SLE, alongside exploring possible associations between specific US findings and clinical and serological markers of SLE disease activity.

## 7. Skin Disease Biomarkers

Cutaneous lupus erythematosus [CLE] refers to skin manifestations which can be limited to the skin or seen in the context of systemic disease. CLE is relatively well characterised in adults, with 5–25% of patients presenting with CLE eventually developing SLE, with a higher risk within the first-year post diagnosis [194]. CLE is less well characterised in children; studies suggested a higher prevalence of systemic manifestations in children with CLE features at disease onset, ranging from 66% [195] to 89% [196].

JSLE patients can have CLE specific manifestations, which include acute, subacute and chronic CLE, as well as non-specific manifestations, which can manifest in clinical settings other than CLE (such as alopecia, purpura, melasma or linear morphea, etc.), posing at times diagnostic challenges [196]. Children with JSLE frequently have mucocutaneous signs and symptoms as part of their initial presentation, and therefore if recognised, they can serve as important diagnostic clues [197]. Mucocutaneous lesions comprise four diagnostic criteria within the revised ACR/EULAR 2019 SLE classification criteria [198], with malar rash considered the most common specific dermatological manifestation in JSLE. Other frequently observed but nonspecific findings include, oral ulcers, photosensitivity, non-scarring alopecia, and vasculitis [195,196,197,199,200].

A large UK national cohort study of JSLE patients identified that individuals who fulfilled the ACR 1997 classification criteria but without any of the mucocutaneous criteria at diagnosis, had an increased risk of major organ involvement, including neurological, renal, and haematological involvement, respectively [201]. However, the presence of a malar rash has been shown in some studies to be suggestive of more severe systemic disease in both adults [202] and children with SLE [196]. 

A retrospective review of 337 patients with adult-onset SLE found that cutaneous manifestations may have prognostic implications related to long term systemic involvement in SLE. Multivariate analysis of a JSLE subgroup revealed an association between acute lupus erythematous and non-scarring alopecia and an increased risk of arthralgia, mucosal ulcers, leukopenia, cutaneous vasculitis and seizures [203]. However, these results have not been validated in other cohort studies. A retrospective study of 40 children with discoid lupus [DL] revealed that 15 children fulfilled the 1982 ACR criteria either concurrently (6/15) or following their DL diagnosis (9/15) within a median follow-up of 5 years [204]. These patients fulfilled the mucocutaneous criteria, haematological or immunological but did not exhibit other end-organ damage, suggesting a possible milder course of JSLE in those patients.

Skin biopsy can be useful tool for CLE diagnosis; however, the biopsy often is not diagnostic, especially in the context of CLE-nonspecific manifestations. Characteristic histopathological and immunofluorescent findings are usually only evident in CLE-specific lesions [205], with immunofluorescence often showing antibody deposition [IgG and/or IgM] and complement components along the dermal-epidermal junction [lupus band test] [206,207]. Despite the good diagnostic yield of skin biopsy in the context of specific CLE lesions, this invasive procedure carry an additional risk of infection and scarring, and not always easy to recommend in children [197]. A recent review on biomarkers for CLE has described the increased expression of type I interferon-related proteins, as well as raised mRNA expression of type 1 interferon-related genes in the skin lesions of patients with CLE compared to control groups, which could be of diagnostic value [208]. Some of these type I interferon-related proteins have also been used to assess treatment response. In a randomized, double-blind, placebo-controlled trial for the drug BIIB059, which is a humanized monoclonal antibody binding to the blood plasmacytoid dendritic cells antigen 2 (BDCA2), the expression of IFN-regulated myxovirus resistance *protein* 1(MxA) was measured using immunohistochemistry at baseline and repeat punch biopsies at week 4. Six of 8 SLE patients who were considered responders, based on the Cutaneous Lupus Erythematosus Disease Area and Severity Index Activity (CLASI-A) score, demonstrated a significant reduction of MxA expression in the skin lesion.

A panel of autoantibodies and proteins were tested in a cohort of SLE patients to identify system-specific markers of disease activity [209]. This study identified that a selection of IgA antibodies were more elevated in patients with DL and there was a correlation between the total IgA level and the skin component of lupus activity index [LAI]. Moreover IL-23 was also higher in these patients and there was a higher expression of IL-23A mRNA in the discoid lesions than in healthy donors’ skin punch biopsy samples.

An autoantibody cluster analysis study found association between malar rash and presence of anti-dsDNA antibodies alone in JSLE [36]. 

Considering the significant impact on patients’ quality of life [210], further research is needed to find alternative non-invasive biomarkers that can predict skin manifestation severity, risk of re-occurrence and association with systemic disease severity and response to available therapies.

## 8. Biomarkers for Haematological Manifestations

Haematological abnormalities are common in individuals with SLE. Leukopenia (with our without lymphopenia), thrombocytopenia, and autoimmune haemolytic anaemia are included in the disease activity scores as well as in various classification criteria for SLE [23,24,211]. However, it is important to be aware that these abnormalities may not be a directly associated with SLE, but instead could be a consequence of SLE treatment, or a manifestation of a separate pathology [24,212]. To our knowledge there are no paediatric studies that have identified biomarkers to predict haematological involvement in JSLE, as such, although there is evidence in a JSLE cluster analysis study that anti-Ro and anti-ribosomal P antibodies correlated with the presence of haemolytic anaemia [36], similarly to older adult SLE studies [213,214]. 

Lymphopenia is extremely common in SLE, and it is historically used as a marker in clinical practice to aid diagnosis and monitor disease activity. However, it is notoriously non-specific, and was subsequently voted to be excluded from the revised ACR/EULAR 2019 classification criteria for the diagnosis of SLE [24]. Nevertheless, lymphopenia should still be monitored, but not overinterpreted in the diagnosis of SLE [24], with some adult and paediatric observational studies suggesting an association between lymphopenia and increased disease activity and organ damage [215,216]. 

Some observational studies found acute thrombocytopaenia to be an important predictor of disease damage [217,218], and a marker of poor prognosis in children with JSLE [217] (Table 5).

## 9. Other Biomarkers

### 9.1. IFN Signature and IFN Associated Proteins

The role of IFN in JSLE and SLE pathogenesis is relatively well established, as various studies showed increased expression of genes related to type I IFN signalling pathway [IFN signature], however the role of type I IFN signature in defining certain subgroups of patients or predicting disease activity is more controversial [219,220,221].

Type 1 IFNs comprise a family of related proteins with immunomodulatory properties acting on the same receptor, the type I IFN receptor (IFNAR) and encoded by 13 different genes clustered on chromosome 9, which have a key role in immune inflammation and anti-viral defence [222]. IFNα subtypes are the most important members of type 1 IFN family and are produced through TLR signaling. TLR 7 is activated single-stranded RNA molecules. SLE is characterised by the production of antibodies against RNA-containing protein complexes such as Sm, RNP, Ro, and La, which subsequently promotes the TLF-7 medicated production of IFN*α* [223]. IFNα contributes to SLE pathogenesis through CD4+ T cell activation in the presence of IFNα producing dendritic cells, while regulatory T cell development is inhibited. In addition, IFNα prevents B cell apoptosis and enhance their proliferation and differentiation into antibody producing plasma cells, therefore leading to abnormal SLE autoimmune activation [224]. 

Studies in children with JSLE found a variable proportion of patients with an increased IFN signature: e.g., 57% in one study [225], and 87% in another [226] compared to age-matched children, however this did not correlate with the disease activity but classified JSLE patients based on their C3 and dsDNA levels [226]. This is somehow similar to more recent adult studies in which the IFN signature reflected a serological signature of SLE without being able to stratify patients based in disease activity scores [227], while other studies correlated IFN signature and its related proteins, such as sialic acid-binding Ig-like lectin 1 (SIGLEC-1) were associated with future flares [228] as well as ethnicity and renal involvement [229]. 

The expression of tetherin, an IFN-induced protein ubiquitously expressed on leukocytes, was investigated as another biomarker of SLE activity in adults. Memory B cell tetherin expression was found to associate with SLE diagnosis and disease activity, as well as predicted future flares [230]. The ex-vivo expression of tetherin on B cells and plasmacytoid dendritic cells was higher in adolescent girls compared to boys, suggesting that it may contribute to the female-bias seen in JSLE [231]. 

Treatment with anifrolumab, a human monoclonal antibody to type I interferon receptor subunit 1 was associated with decreased IFN signature in adult-onset SLE and clinical benefit in a phase 3 RCT [232], despite initially not meeting the primary endpoint, SLE responder index 4 (SRI4) [233]. Future research is required to assess this treatment efficacy in JSLE. 

### 9.2. Long Non-Coding and Micro RNAs

Long non-coding RNA (lncRNA) and microRNA (miRNA) are non-protein-coding RNAs involved in the regulation of various biologic processes, including chronic inflammation, degenerative and metabolic disease and malignancy [234], through regulation of gene transcription and protein functions.

Recent research showed that lncRNAs are likely to be associated with cell differentiation and activation and play an important regulatory role in the differentiation and activation of immune cells in congenital and acquired immune conditions, including SLE [235]. 

MiRNAs are evolutionally conserved single-stranded noncoding RNAs with role in the post-transcriptional regulation of gene expression and RNA silencing as they negatively regulate gene expression at the messenger RNA (mRNA) and protein level. The control of miRNA mediated gene expression is critical for normal cellular functions, such as cell cycle, differentiation, apoptosis, metabolism among others, and therefore involved in maintaining the normal immune system development and function, as well as in the risk of developing SLE [236]. 

Although data about their biomarker performance in JSLE is limited, both lncRNA and miRNA were differentially expressed in JSLE patients compared to age matched healthy children [237], suggesting their contribution to JSLE pathogenesis.

Several studies reported their role in the non-invasive diagnosis of SLE, providing evidence for their ability to differentiate with high accuracy SLE patients from patients with other ARDs [238]. In addition, various lncRNAs have been investigated as biomarkers for SLE nephritis in adults; the expression of certain lncRNAs was increased in kidney biopsies from LN patients and positively correlated with disease activity and interferon [IFN] scores [239], while the neutrophil lncRNA expression profile was validated as predictor marker for development of nephritis in SLE patients [240]. 

## 10. Discussion

JSLE is a complex ARD with heterogeneous clinical manifestations and unpredictable outcome, requiring tailored management strategies. Recent research efforts have been directed towards identifying biomarkers with various clinical uses in a quest to improve the recognition, diagnosis, prognosis as well as personalised management approaches in adult-onset SLE, with some evidence of emerging from JSLE studies. This narrative review summarises the literature supporting the validity and potential clinical applications of many JSLE biomarkers, comprising a large array of clinical tests (e.g., PedANAM test in NPSLE), serum biomarkers (e.g., ApoB:ApoA1 ratio in CVD), urinary biomarkers (e.g., urinary NGAL in LN), combination biomarker panels (e.g., AGP-1/ceruloplamin/L-PGDS/transferrin in LN), tissue biopsy (e.g., renal biopsy in LN) and radiological biomarkers (US in arthritis). 

Clinical, radiological and tissue specific biomarkers of JSLE specific organ involvement require adequate training, can be time-consuming to perform and interpret and, if performed regularly and on a large scale, may prove expensive. Additionally, clinical and radiological assessments are less objective due to inter-observer variability. Serum and urinary biomarkers, however, do not require trained clinicians and, if reproducible assays are made available outside of a research setting, they may prove cheaper and more scalable. Additionally, laboratory quantifiable biomarkers if well standardised could have the advantage of being more objective. Urinary biomarkers are a particularly promising as they may potentially reduce the need for invasive investigations such as blood tests and renal biopsies in a young patient population experiencing a heavy burden of medical care from early age. However, while some biomarkers are promising in terms of accessibility and cost, the evidence for their clinical utility is still limited. In addition, some reflect only specific disease manifestations and cannot be used widely (e.g., there is only evidence for a role for urinary biomarkers in JSLE nephritis assessment, while they do not correlate with extrarenal disease activity).

There is a large amount of variation in the level of evidence related to the validity and sensitivity to change of various biomarkers tested in JSLE. The evidence for potential clinical utility is strongest for renal biomarkers, which have been most widely studied, with some externally validated in multiple JSLE cohorts. For example, there have been nine studies that have investigated the role of the urinary biomarker NGAL, including a large sample size of 581 JSLE patients. However, most of these studies have been focused on UK and US cohorts, with a few studies including Egyptian [57,58] and South African [59,60] JSLE patients; therefore the impact of genetic background and ethnicity has not been evaluated. 

The evidence is less strong for the utility of biomarkers for extra-renal JSLE, as they have been investigated in fewer studies, with smaller sample size and only few have been validated in other JSLE cohorts. There are multiple possible explanations for this. LN is the most common, severe organ manifestation in JSLE, occurring in up to half of children [6], whereas other organ involvement (especially CNS) is rarer, or poses less clinical challenges in terms of morbidity and mortality risk (e.g., arthritis and cutaneous manifestations), although significantly impacting patients’ quality of life. Furthermore, there is higher disease heterogenicity in NPSLE compared to LN, and the absence of a ‘gold-standard’ diagnostic test in NPSLE compared to the renal biopsy in LN impacts the ability to design good quality studies with large sample size. Various JSLE biomarkers are associated with different degrees of technical challenges, cost and accessibility (such as carotid ultrasound, fMRI, MSUS), which limits their widespread use.

The limitations in the current use of organ-specific biomarkers in clinical and therapeutic decisions in JSLE are related to the lack of cross-validation across ethically and racially diverse cohorts, as well as difficulty in standardising the assays used for their detection across the globe. For example, significant differences in the sensitivity of various assays used for ANA and anti-ds-DNA antibody detection exist [241], which prompted the suggestion of implementation of computer-aided automated immunofluorescence analysis [242]. Further research into improving methodological aspects of biomarker detection is required.

## 11. Conclusions

This review identified numerous biomarkers for organ involvement in JSLE, suggesting a substantial research investment in a better understanding and management of this rare disease, which is still associated with increased morbidity and mortality and an unmet need for better patient management and improved long-term outcomes. Evaluating biomarkers associated with other JSLE co-morbidities and complications (such as bone health, puberty-related outcomes, obesity, response to certain therapies used in JSLE, treatment related toxicity or JSLE damage) was beyond the purpose of this review. Although the evidence for biomarker clinical utility in large cohort studies with long-term follow-up is lacking, the progress achieved in identifying and validating biomarkers for JSLE has been significant. 

As far as LN is concerned, further research is needed to validate biomarkers in international multi-ethnic cohorts and to establish their role in the pathogenesis of disease, sensitivity to change overtime as well as their potential as therapeutic targets. Long-term follow-up studies are required, in particular for CVD biomarkers, as the clinical significance of accelerated atherosclerosis may not be apparent for years.

It is unlikely that a single ‘gold standard’ biomarker will be proven as clinically useful in assessing JSLE organ involvement; instead, different biomarkers, or possibly composite biomarker panels like those suggested for assessment of LN, may be used in various clinical scenarios. Highly sensitive biomarkers will be needed for early detection of organ involvement, but highly specific biomarkers are needed to confirm the diagnosis. Biomarkers that can predict JSLE severity and response to treatment are needed, so that patients can be stratified according to risk and adequate therapeutic strategy. Sensitive biomarkers can help evaluating the disease dynamic and progression over time, as well as guide treatment optimisation as required. Additionally, biomarkers suitable for organ damage detection in the absence of active disease are required to facilitate differentiation between various disease states. Further experience derived from real life clinical practice will drive the selection of the best biomarker candidates based on cost, speed, ease of use, availability and reliability across different SLE populations and ages.

## Figures and Tables

**Figure 1 ijms-22-07619-f001:**
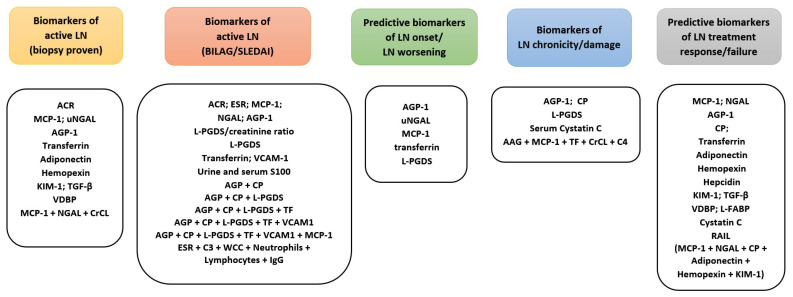
Biomarkers with potential clinical utility in JSLE associated LN. AAG α(1) -acid glycoprotein, AGP-1 Alpha-1-acid Glycoprotein, AGP-1 Alpha-1-acid Glycoprotein, BILAG British Isles Lupus Assessment Group, C4 Complement fraction 4, CP Ceruloplasmin, CrCl creatinine clearance, KIM-1 Kidney Injury Molecule-1, L-FABP Liver-Fatty Acid Binding Protein, LN lupus nephritis, L-PGDS Lipocalin-type Prostaglandin-D Synthetase, MCP-1 Monocyte Chemoattractant Protein-1, uNGAL urinary Neutrophil Gelatinase Associated Lipocalin, RAIL Renal Activity Index in Lupus, SLEDAI Systemic Lupus Erythematosus Disease Activity Index, TGFβ Transforming Growth Factor β, TF transferrin, u urinary, VCAM-1 Vascular cell adhesion protein 1, VDBP Vitamin-D Binding Protein, WCC white cell count.

**Table 1 ijms-22-07619-t001:** Classical renal biomarkers.

Biomarker	Study	Definition of Active Nephritis	Patient Number	Findings
C3/C4	Smith 2017 [44] Watson 2013 [45] Smith 2017 [46] Brunner 2012 [47] Brunner 2016 [48] Suzuki 2009 [49] Watson 2012 [50]	BILAG BILAG BILAG SLEDAI, Biopsy (BAI) Biopsy (NIH-AI and TIAI) SLEDAI, BILAG BILAG	91 64 370 76 47 98 60	No significant difference in C3 levels in active LN in the UK and US cohorts. C3 is a poor predictor of active LN (AUC 0.65 UK, 0.64 US). C4 was not a predictor of active LN (AUC 0.59 UK, 0.48 US). C3 and C4 significantly lower in active LN (*p* < 0.001 and *p* 0.009, respectively). C3 is an independent predictor of active LN (β −0.459, CI −0.663 to 0.254, *p* < 0.001). Negative correlation between MCP-1 and serum C3 (Rho −0.22, *p* = 0.002). C3 and C4 significantly lower in active LN (*p* < 0.001 and *p* = 0.017, respectively). C3 weakly correlated with SLEDAI LN (r −0.34, *p* < 0.0043). C4 not correlated with SLEDAI LN (r < 0.2). No significant difference in C3/C4 levels in active BAI LN. C3 and C4 were significantly lower in active NIH-AI LN (*p* = 0.005 and 0.015, respectively) but not in active TIAI LN (*p* = 0.931 and 0.981, respectively). No significant difference in C3 or C4 levels in active LN (SLEDAI or BILAG). C3 was not a predictor of active SLEDAI LN (AUC 0.58), was a poor predictor of active BILAG LN (AUC 0.63) and a good predictor of SDI LN chronicity (AUC 0.75). C4 was a poor predictor of active SLEDAI LN (AUC 0.60), not a predictor of active BILAG LN (AUC 0.49) and a poor predictor of SDI LN chronicity (AUC 0.64). Significantly lower in active LN (*p* = 0.02).
dsDNA	Smith 2017 [44] Watson 2014 [45] Smith 2017 [46] Brunner 2016 [48] Watson 2012 [50]	BILAG BILAG BILAG Biopsy (NIH-AI and TIAI) BILAG	91 64 370 47 60	No significant difference in active LN in the UK and US cohorts. Poor predictor of active LN (AUC 0.62 UK, 0.64 US). Significantly higher in active renal disease (*p* < 0.001). Not a predictor of worsening LN (AUC 0.44; *p* = 0.62). No significant difference in active LN (*p* = 0.068). No significant difference in active NIH-AI or TIAI LN (*p* = 0.353 and 0.849, respectively). No significant difference in active LN (*p* = 0.34).
Proteinuria	Smith 2017 [44] Watson 2014 [45] Smith 2017 [46] Brunner 2012 [48] Brunner 2016 [48] Suzuki 2009 [49] Watson 2012 [50] Smith 2018 [51]	BILAG BILAG BILAG SLEDAI, Biopsy (BAI) Biopsy (NIH-AI and TIAI) SLEDAI, BILAG BILAG BILAG, Biopsy	91 64 370 76 47 98 60 64	uACR significantly higher in active LN in the UK and US cohorts (*p* < 0.01 and *p* = 0.03, respectively). uACR significantly higher in active LN (*p* < 0.001). Not a predictor of worsening LN (AUC 0.46; *p* = 0.76). uACR and uPCR significantly higher in active LN (both *p* < 0.001). uPCR weakly correlated with SLEDAI LN (r = 0.40, *p* < 0.0004). Significantly higher in active BAI LN (*p* = 0.008). Good predictor of active BAI LN (AUC 0.76). uPCR was not significantly different in active LN. uPCR significantly higher in active SLEDAI and BILAG LN (both *p* < 0.0001). Excellent predictor of SLEDAI LN activity (AUC 0.91), a good predictor of BILAG LN activity (AUC 0.85), and a fair predictor of SDI LN chronicity (AUC 0.76). uACR significantly higher in active LN (*p* < 0.001). An elevated uACR or uPCR persists after active LN flare in 61% of patients. Age at time of LN flare (*p* = 0.007), eGFR (*p* = 0.035), and haematological involvement at the time of LN onset (*p* = 0.016) were significantly associated with time to normalisation of uACR and uPCR.
Creatinine	Smith 2017 [44] Watson 2014 [45] Smith 2017 [46] Brunner 2012 [47]	BILAG BILAG BILAG SLEDAI, Biopsy (BAI)	91 64 370 76	No significant difference in active LN in the UK and US cohorts. Significantly higher in active LN (*p* = 0.026). Poor predictor of worsening LN (AUC 0.52; *p* = 0.79). No significant difference in active LN (*p* = 1.0). Weakly correlated with SLEDAI LN (r = 0.29, *p* < 0.01). No significant difference in active BAI LN.
eGFR	Smith 2017 [44] Watson 2013 [45] Smith 2017 [46] Brunner 2012 [47] Brunner 2016 [48] Suzuki 2009 [49] Watson 2012 [50]	BILAG BILAG BILAG SLEDAI, Biopsy (BAI) Biopsy (NIH-AI and TIAI) SLEDAI, BILAG BILAG	91 64 370 76 47 98 60	No significant difference in active LN in the UK and US cohorts. No significant difference in active LN (*p* = 0.085). Poor predictor of worsening LN (AUC 0.54; *p* = 0.75) No significant difference in active LN (*p* = 1.0). No correlation with SLEDAI LN (r < 0.2). No significant difference in active BAI LN. Significantly higher in active NIH-AI and TIAI LN (*p* = 0.003). No significant difference in active LN. Not a predictor of SLEDAI or BILAG LN activity (AUC 0.45 and 0.50, respectively). Not a predictor of SDI LN chronicity (AUC 0.39). No significant difference in active LN (*p* = 0.58).
CRP	Smith 2017 [46]	BILAG	370	No significant difference in active LN (*p* = 1.0).
ESR	Smith 2017 [44] Watson 2013 [45] Smith 2017 [46] Watson 2012 [50]	BILAG BILAG BILAG BILAG	91 64 370 60	Significantly higher in active LN in the UK cohort (*p* < 0.01). Fair predictor of active LN (AUC 0.796) in the UK cohort. Not measured in the US cohort. Significantly higher in active LN (*p* 0.018). Not a predictor of worsening LN (AUC 0.42; *p* 0.52). Significantly higher in active LN (*p* < 0.001). No significant difference in active LN (*p* = 0.07).

Legend: AUC, area under curve; BAI, Biopsy Activity Index; BILAG, British Isles Lupus Assessment Group; CRP, C-reactive protein; dsDNA, double-stranded DNA; eGFR, estimated glomerular filtration rate; ESR, erythrocyte sedimentation rate; LN, lupus nephritis; MCP-1, monocyte chemoattractant protein-1; SDI, Systemic Lupus International Collaborating Clinics/American College of Rheumatology damage index; SLEDAI, Systemic Lupus Erythematosus Disease Activity Index; NIH-AI, National Institutes of Health-Activity Index; TIAI, Tubulointerstitial Activity Index; uACR, urinary albumin/creatinine ratio; uPCR, urinary protein/creatinine ratio.

**Table 2 ijms-22-07619-t002:** Validated novel urinary renal biomarkers.

Biomarker	Study	Definition of Active Nephritis	Patient Number	Findings
MCP-1	Smith 2017 [44] Watson 2014 [45] Brunner 2012 [47] Brunner 2016 [48] Watson 2012 [50] Brunner 2017 [54]	BILAG BILAG SLEDAI, Biopsy (BAI) Biopsy (NIH-AI andTIAI) BILAG SLEDAI	91 64 76 47 60 87	Significantly higher in active LN in the UK (*p* = 0.028) and US cohorts (*p* < 0.001). MCP-1 and MCP-1-crea are significantly higher in active LN (*p* < 0.001 and *p* = 0.006, respectively). An independent predictor of active LN (β 0.183, CI 0.60–0.305, *p* = 0.004). MCP-1 is a good predictor of improved renal disease over time (AUC 0.81, *p* = 0.013). Good predictor of BAI activity (AUC 0.82). Weakly correlated with SLEDAI LN activity (r = 0.23, *p* < 0.07). Significantly higher in presence of mesangial proliferation (*p* 0.008) and capillary proliferation (*p* = 0.014) on biopsy. Significantly higher in active LN on NIH-AI (*p* = 0.000) + TIAI (*p* = 0.035). Significantly higher in biopsies which had endocapillary hypercellularity, leukocyte infiltrates, subendothelial deposits, tubular cell flattening and tubular cell necrosis (all *p* < 0.05). Fair predictor for NIH-AI LN activity (AUC > 0.7). Significantly higher in active LN (*p* = 0.005). At 3 and 6 months after therapy, MCP-1 was significantly higher in non-responders to therapy compared to responders (*p* < 0.05 and *p* < 0.005, respectively), although there was no significant difference at baseline.
NGAL	Smith 2017 [44] Watson 2014 [45] Brunner 2016 [48] Suzuki 2008 [49] Hinze 2009 [55] Brunner 2006 [56] Gheita 2015 [57] Hammad 2013 [58] Brunner 2017 [54]	BILAG BILAG Biopsy (NIH-AI and TIAI), SLEDAI BILAG, SLEDAI, PA SLEDAI, biopsy Biopsy SLEDAI SLEDAI BILAG	91 64 47 85 111 35 28 33 87	No significant difference in active LN in the UK and US cohorts (*p* = 1.0) NGAL and NGAL-crea are significantly higher in active LN (*p* = 0.001 and 0.02, respectively). Fair predictor of worsened renal disease activity over time (AUC 0.76, *p* = 0.04). Significantly higher in NIH-AI active LN (*p* = 0.017) but not TIAI active LN (*p* = 0.094). Significantly higher in biopsies which had endocapillary hypercellularity (*p* < 0.05) and epithelial cells in tubular lumen (*p* < 0.05). Fair predictor of NIH-AI LN activity (AUC > 0.7). NGAL-crea was significantly higher in active LN (*p* = 0.02) and weakly correlated with LN activity (r = 0.4, *p* < 0.008). Worsening LN was associated with an increase in NGAL (380%) and NGAL-crea (125%) (*p* < 0.01). Significant increase in NGAL 3–6 months before flare of LN (+104% BILAG [*p* = 0.01] +70% SLEDAI [*p* = 0.03], +70% PA [*p* = 0.04]). Good predictor of BILAG LN activity (AUC 0.8 [sensitivity 81.8%, specificity 82.4%, PPV 60.7%, NPV 93.2%]), and fair predictor of SLEDAI LN activity (AUC 0.78 [sensitivity 82.6%, specificity 70.9%, PPV 48.6%, NPV 92.4%]). NGAL-crea was significantly higher in active LN (*p* < 0.0005) and moderately correlated with LN activity (r > 0.59 *p* < 0.0001). NGAL-crea was weakly correlated with SDI LN chronicity (r >0.47 *p* < 0.001). NGAL-crea was 16-fold higher in biopsy-proven LN than in those without LN (*p* = 0.0002) and strongly correlated with biopsy LN activity (r = 0.73, *p* < 0.001) and chronicity (r > 0.58 *p* = 0.004). Excellent predictor of biopsy-proven LN (AUC 0.944 [sensitivity 90%, specificity 100%]). Significantly higher in patients with LN on biopsy (*p* = 0.019) and weakly correlated with the presence of LN (r = 0.3 *p* = 0.02). No significant difference in NGAL in active LN. However, NGAL-crea was significantly higher in active LN (*p* < 0.001) and moderately correlated with LN activity (r = 0.5 *p* = 0.02). NGAL was significantly predictive of proliferative LN (class III and IV) (*p* = 0.005, 91% sensitivity, 70% specificity. NGAL was significantly higher in non-responders to therapy compared to responders (*p* < 0.005) 6 months after therapy, although there was no significant difference at baseline and at 3 months.
AGP-1	Smith 2017 [44] Brunner 2012 [47] Brunner 2016 [48] Suzuki 2009 [49] Watson 2012 [50] Smith 2019 [59] Smith 2018 [60] Brunner 2017 [54]	BILAG SLEDAI Biopsy (BAI) Biopsy (NIH-AI and TIAI) SLEDAI, BILAG BILAG BILAG BILAG SLEDAI	91 76 47 98 60 80 23 87	Significantly higher in active LN in the UK (*p* < 0.001) and US cohorts (*p* < 0.001). Weakly correlated with SLEDAI LN activity (r = 0.35, *p* < 0.003). Fair predictor of BAI activity (AUC 0.76). Significantly higher when mesangial proliferation (*p* < 0.005) and cellular crescents (*p* < 0.003) present on biopsy. Significantly higher in active TIAI LN (*p* = 0.043) but not NIH-AI active LN (*p* = 0.377). Significantly higher when tubular cell necrosis (*p* < 0.05) and macrophages in tubular lumen (*p* < 0.05) present on biopsy. Urinary AGP-1: Significantly higher in JSLE vs. JIA (*p* < 0.0001). Significantly higher in active SLEDAI (*p* = 0.005) and BILAG (*p* < 0.0001) LN. uAGP-1 significantly increased at least 3 months before diagnosis of worsening LN activity (*p* < 0.009). uAGP-1-crea was significantly higher in active LN (*p* < 0.0001) but there was no significant difference in uAGP-1-prot. uAGP-1 is a fair predictor of SLEDAI LN activity (AUC 0.76), a good predictor of BILAG LN activity (AUC 0.81) and a fair predictor of SDI LN damage (AUC 0.73). Serum AGP-1: No difference in JSLE vs. JIA. No difference in active LN. Significantly higher in active LN (*p* = 0.027). Predictive of LN flare 3, 9 and 12 months before (HR 1.49 [95% CI 1.10–2.02]). Best predictor of LN activity on Markov Multi-State model (AICc 139.81). Significantly higher in active LN (*p* < 0.001). AGP-1 was significantly higher in non-responders to therapy compared to responders at baseline (*p* = 0.023) and 3, 6 and 12 months after therapy (*p* < 0.005 for all). Excellent predictor of non-responders to therapy at 3 months (AUC 0.98).
CP	Smith 2017 [44] Brunner 2016 [48] Suzuki 2009 [49] Smith 2019 [59] Smith 2018 [60] Brunner 2017 [54]	BILAG Biopsy (NIH-AI and TIA) SLEDAI, BILAG BILAG BILAG SLEDAI	91 47 98 80 23 87	Significantly higher in active LN in the UK (*p* < 0.001) and US cohorts (*p* < 0.001). Significantly higher in active NIH-AI LN (*p* = 0.015) but not active TIAI LN (*p* = 0.138). Urinary CP: Significantly higher in JSLE vs. JIA (*p* < 0.0001) and significantly higher in active SLEDAI (*p* = 0.004) and BILAG (*p* = 0.003) LN. There was no significance in uCP-crea or uCP-prot in active LN. uCP is a poor predictor of SLEDAI LN activity (AUC 0.68), a good predictor of BILAG LN activity (AUC 0.80) and a good predictor of SDI LN damage (AUC 0.84). Serum CP: No difference in JSLE vs. JIA. No difference in active LN. Predictive of LN remission 3, 9 and 12 months before (HR 0.60 [95% CI 0.39–0.93]. Second best predictor of LN activity on Markov Multi-State model after AGP-1 (AICc 141.40). Significantly higher in active LN (*p* < 0.001) CP was significantly higher in non-responders to therapy compared to responders at baseline (*p* = 0.006), 3 months (*p* < 0.005) and 6 months after therapy (*p* < 0.005). Good predictor of non-responders to therapy at 3 months (AUC 0.83).
L-PGDS	Smith 2017 [44] Brunner 2012 [47] Brunner 2016 [48] Suzuki 2009 [49] Smith 2018 [60] Brunner 2017 [54]	BILAG SLEDAI Biopsy (BAI) SLEDAI, BILAG BILAG SLEDAI	91 76 47 98 23 87	Significantly higher in active LN in the UK (*p* < 0.001) and US cohorts (*p* = 0.021). Weakly correlated with SLEDAI LN activity (r = 0.28, *p* < 0.016). Not associated with any histological features. No significant difference based on NIH-AI (*p* = 0.061) or TIAI (*p* = 0.081) LN activity. Significantly higher when endocapillary hypercellularity present on biopsy (*p* < 0.05). Urinary L-PGDS: Significantly higher in JSLE vs. JIA (*p* < 0.0025) and significantly higher in active SLEDAI (*p* < 0.0001) and BILAG (*p* = 0.004) LN. uL-PGDS significantly increased at least 3 months before clinical diagnosis of worsening LN activity (*p* < 0.009). L-PGDS-crea was significantly higher in active LN (*p* < 0.0005) but L-PGDS-protein was significantly lower in active LN (*p* < 0.009). uL-PGDS is a fair predictor of SLEDAI LN activity (AUC 0.71) and BILAG LN activity (AUC 0.73) and a good predictor of SDI LN damage (AUC 0.84). Serum L-PGDS: No difference in JSLE vs. JIA. No difference in active LN. Significantly higher in active LN (*p* = 0.018). L-PGDS was significantly higher in non-responders to therapy compared to responders at baseline (*p* = 0.044) and 3 months (*p* < 0.05). Excellent predictor of non-responders to therapy at 3 months (AUC 0.96).
Transferrin	Smith 2017 [44] Brunner 2012 [47] Brunner 2016 [48] Suzuki 2009 [49] Smith 2018 [60] Brunner 2017 [54]	BILAG SLEDAI, Biopsy (BAI) Biopsy (NIH-AI and TIA) SLEDAI, BILAG BILAG SLEDAI	91 76 47 98 23 87	No significant difference in active LN in the UK cohort (*p* = 0.063), significantly higher in active LN group in the US cohort (*p* < 0.001). No correlation to SLEDAI LN activity (r < 0.2). Fair predictor of BAI LN activity (AUC 0.76). Significantly higher when mesangial proliferation (*p* = 0.024), capillary proliferation (*p* = 0.017) and cellular crescents present on biopsy (*p* = 0.024) Significantly higher in NIH-AI active LN (*p* = 0.029) but not in TIA active LN (*p* = 0.478) Urinary TF: Significantly higher in JSLE vs. JIA (*p* ≤ 0.0001) and significantly higher in active SLEDAI (*p* < 0.0001) and BILAG (*p* < 0.0001) LN. Levels significantly increased at least 3 months before clinical diagnosis of active flare (*p* < 0.0009). uTF-crea and uTF-prot were significantly higher in active LN (*p* <0.0001 and *p* < 0.05, respectively). uTF is a good predictor of SLEDAI LN activity (AUC 0.80), BILAG LN activity (AUC 0.81) and SDI LN damage (AUC 0.84). Serum TF: No difference in JSLE vs. JIA. No difference in active LN. Significantly higher in active LN than inactive LN (*p* < 0.05). Transferrin was significantly higher in non-responders to therapy compared to responders at baseline (*p* = 0.012) and 3 months, 6 and 12 months (*p* < 0.05). Excellent predictor of non-responders to therapy at 3 months (AUC 0.95).

Legend: AICc, Akaike Information Criterion; AGP-1, alpha-1-acid glycoprotein; AUC, area under curve; BAI, Biopsy Activity Index; BILAG, British Isles Lupus Assessment Group; CI, confidence interval; CP, ceruloplasmin; JIA, juvenile idiopathic arthritis; LN, lupus nephritis; L-PGDS, lipocalin-type prostaglandin-D synthetase; MCP-1, monocyte chemoattractant protein-1; MCP-1-crea, monocyte chemoattractant protein-1/creatinine ratio; NGAL, neutrophil gelatinase-associated lipocalin; NGAL-crea, neutrophil gelatinase-associated lipocalin to creatinine ratio; NIH-AI, National Institutes of Health-Activity Index; PA, physician assessment of disease activity; SDI, Systemic Lupus International Collaborating Clinics/American College of Rheumatology damage index; SLEDAI, Systemic Lupus Erythematosus Disease Activity Index; TIAI, Tubulointerstitial Activity Index; TF, transferrin; uAGP-1, urine alpha-1-acid glycoprotein; uAGP-1-crea, urine alpha-1-acid glycoprotein/creatinine ratio; uAGP-1-prot, urine alpha-1-acid glycoprotein/protein ratio; uCP, urine ceruloplasmin; uCP-crea, urine ceruloplasmin/creatinine ratio; uCP-prot, urine ceruloplasmin/protein ratio; uL-PGDS, urinary lipocalin-type prostaglandin-D synthetase; uL-PGDS-crea, urinary lipocalin-type prostaglandin-D synthetase/creatinine ratio; uL-PGDS-prot, urinary lipocalin-type prostaglandin-D synthetase/protein ratio; uTF-crea, transferrin/creatinine ratio; uTF-protein transferrin/protein ratio.

**Table 3 ijms-22-07619-t003:** Other novel renal biomarkers investigated.

Biomarker	Study	Definition of Active Nephritis	Patient Number	Findings
VCAM-1	Smith 2017 [44] Smith 2018 [60]	BILAG BILAG	91 23	Significantly higher in active LN in the UK (*p* = 0.007) and US cohorts (*p* < 0.001). Significantly higher in active LN (*p* = 0.010).
Adiponectin	Brunner 2016 [48] Brunner 2017 [54]	Biopsy SLEDAI	47 87	Significantly higher in NIH-AI and TIAI active LN (*p* = 0.023 and 0.024, respectively). Significantly higher in non-responders to therapy compared to responders at 3, 6 and 9 months (*p* < 0.05). Excellent predictor of non-responders to therapy at 3 months (AUC 0.90).
Hemopexin	Brunner 2016 [48] Brunner 2017 [54]	Biopsy SLEDAI	47 87	Significantly higher in TIAI active LN (p 0.010) but not in NIH-AI active LN (*p* = 0.138). Significantly higher in non-responders to therapy compared to responders at 3 and 6 months (*p* < 0.01). Good predictor of non-responders to therapy at 3 months (AUC 0.8).
Hepcidin	Brunner 2016 [48] Brunner 2017 [54]	Biopsy SLEDAI	47 87	No significant difference in NIH-AI and TIAI active LN (*p* = 0.753 and 0.802, respectively). Significantly higher in non-responders to therapy compared to responders at baseline (*p* = 0.037).
KIM-1	Brunner 2016 [48] Brunner 2017 [54]	Biopsy SLEDAI	47 87	Significantly higher in NIH-AI active LN (p 0.000) but not TIAI active LN (*p* = 0.140). Significantly higher in non-responders to therapy compared to responders at 3, 6 and 9 months (*p* < 0.05, *p* < 0.05, and *p* < 0.05, respectively). Good predictor of non-responders to therapy at 3 months (AUC 0.8).
TGF-β	Brunner 2016 [48] Brunner 2017 [54]	Biopsy SLEDAI	47 87	Significantly higher in NIH-AI and TIAI active LN (*p* = 0.013 and 0.030, respectively). Significantly higher in non-responders to therapy compared to responders at 3, 6 and 9 months (*p* < 0.005, *p* < 0.05, and *p* < 0.01, respectively). Good predictor of non-responders to therapy at 3 months (AUC 0.81).
VDBP	Brunner 2016 [48] Brunner 2017 [54]	Biopsy SLEDAI	47 87	Significantly higher in TIAI active LN (*p* = 0.844) but not NIH-AI active LN (*p* = 0.030). Significantly higher in non-responders to therapy compared to responders at baseline, 3, 6 and 12 months (*p* = 0.027, *p* < 0.005, *p* < 0.005, and *p* < 0.05, respectively). Excellent predictor of non-responder to therapy at 3 months (AUC 0.9).
L-FABP	Brunner 2016 [48] Brunner 2017 [54]	Biopsy SLEDAI	47 87	No significant difference in NIH-AI and TIAI active LN (*p* = 0.454 and 0.099, respectively). Significantly higher in non-responders to therapy compared to responders at baseline, 3, 6 and 12 months (*p* < 0.05, *p* < 0.005, and *p* < 0.05, respectively). Good predictor of non-responders to therapy at 3 months (AUC 0.8).
Cystatin C	Brunner 2016 [48] Brunner 2017 [54] Gheita 2015 [61]	Biopsy SLEDAI Biopsy	47 87 28	Urine: No significant difference in NIH-AI and TIAI active LN (*p* = 0.352 and = 0.138, respectively). Urine: Significantly higher in non-responders to therapy compared to responders at 6 months (*p* < 0.05). Fair predictor of non-responders at 3 months (AUC 0.7). Serum: Weakly correlated with SDI LN damage (r = 0.38, *p* 0.003).
Osteopontin	Brunner 2017 [54]	SLEDAI	87	No significant difference in non-responders to therapy compared to responders.
S100	Donohue 2020 [93] Turnier 2017 [94]	BILAG SLEDAI	64 96	Serum: Significantly higher levels of S100A8/A9 (*p* < 0.001) and S100A12 (*p* = 0.035) in JSLE patients when compared to healthy controls. Significantly higher levels of S100A8/A9 (*p* = 0.043) and S100A12 (*p* = 0.021) in JSLE patients with active LN compared to those with inactive/no renal disease. No differences in levels of S100A4. Urine: Significantly higher levels of S100A12 (*p* = 0.029) in JSLE patients when compared to healthy controls. Significantly higher levels of S00A12 (*p* = 0.0095) in JSLE patients with active LN compared to those with inactive/no renal disease. No difference in levels of S100A4 or S100A8/A9. Serum: No significant difference in S100A4, S100A6, S100A8/9 and S100A12 in active LN. Urine: significantly higher levels of S100A4 (*p* < 0.0001), S100A6 (*p* = 0.0008), S100A8/9 (*p* = 0.0235) and S100A12 (*p* = 0.018) in active LN. Levels of all S100 decreased with improvement of LN; S100A4 (*p* < 0.0001), S100A6 (*p* = 0.0075), S100A8/9 (*p* = 0.019), and S100A12 (*p* = 0.0143). S100A4 was higher in proliferative (class III/IV) LN than in membranous (class V) LN (*p* = 0.03).
IP-10	Watson 2012 [50]	BILAG	60	No significant difference in active LN (*p* = 0.55) or between JSLE vs. healthy controls (*p* = 0.13).
sCD25	Hassan 2021 [95]	SLEDAI, SLICC renal activity score	53	Urine: JSLE patients had higher normalised sCD25 levels compared to the healthy controls (*p* = 0.001). JSLE patients with active LN had significantly higher sCD25 levels than active JSLE patients without LN (*p* = 0.002) and JSLE patients with inactive disease (*p* < 0.001). Serum: CD25 levels were significantly higher in active SLEDAI LN (*p* < 0.001, AUC 0.88 [sensitivity 68.7%, specificity 91.9%]), and active SLICC renal activity score LN (*p* < 0.001).
NK cells	Zahran 2021 [96]	Biopsy	35	Peripheral blood: NK strongly negatively correlation with LN activity (r −0.6, *p* = 0.001). CD56^bright^ NK cells strongly correlated with LN activity (r = 0.6, *p* < 0.0001) and moderately correlated with LN chronicity (r = 0.4 *p* = 0.01). NKT moderately negative correlation with LN activity (r = −0.5, *p* = 0.001)

Legend: AUC, area under curve; BILAG, British Isles Lupus Assessment Group; IP-10, interferon gamma-induced protein 10; JIA, juvenile idiopathic arthritis; KIM-1, Kidney Injury Molecule-1; L-FABP, liver-type fatty acid binding protein ; LN, lupus nephritis; NIH-AI, National Institutes of Health-Activity Index; NK, Natural Killer; NKT, Natural Killer T lymphocyte; SDI, Systemic Lupus International Collaborating Clinics/American College of Rheumatology damage index; sCD25, soluble alpha chain of interleukin 2 (CD25); SLEDAI, Systemic Lupus Erythematosus Disease Activity Index; SLICC, Systemic Lupus International Collaborating Clinics; TGFβ, Transforming Growth Factor β; TIAI, Tubulointerstitial Activity Index; VCAM-1, vascular cell adhesion molecule-1; VDBP, vitamin D-binding protein.

**Table 4 ijms-22-07619-t004:** Biomarker combination panels in jSLE.

Biomarker	Study	Findings
AGP-1 + CP AGP-1 + CP + L-PGDS AGP-1 + CP + L-PGDS + TF AGP-1 + CP + L-PGDS + TF + VCAM1 AGP-1 + CP + L-PGDS + TF + VCAM1 + MCP-1 AGP-1 + CP + MCP-1 + PCR AGP-1 + MCP-1 + TF + CrCL + C4 MCP-1 + NGAL + CrCL RAIL (MCP-1 + NGAL + CP + adiponectin + hemopexin + KIM-1) RAIL (MCP-1 + NGAL + CP + adiponectin + hemopexin + KIM-1)- creatinine adjusted ESR + C3 + WCC + Neutrophils + Lymphocytes + IgG	Smith 2017 [44] Smith 2018 [60] Smith 2017 [44] Smith 2018 [60] Smith 2017 [44] Smith 2018 [60] Smith 2017 [44] Smith 2017 [44] Brunner 2012 [47] Brunner 2012 [47] Brunner 2012 [47] Brunner 2016 [48] Brunner 2017 [54] Brunner 2017 [54] Smith 2017 [46]	Good predictor of active BILAG LN in UK (AUC = 0.881) cohort and excellent in US cohort (0.982) Excellent predictor of active BILAG LN (AUC = 0.992 [95% CI = 0.970-1]) Excellent predictor of active BILAG LN in UK (AUC = 0.900) and US cohort (0.982) Excellent predictor of active BILAG LN (AUC 0.992 [95% CI = 0.970-1]) Excellent predictor of active BILAG LN in UK (AUC = 0.920) and US cohort (0.991) Excellent predictor of active BILAG LN (AUC = 1 [95% CI = 1-1]) Excellent predictor of active BILAG LN in UK (AUC = 0.920) and US cohort (0.987) Excellent predictor of active BILAG LN in UK cohort (AUC = 0.920) Good predictor of BAI LN activity (AUC = 0.85 [sensitivity 72%, specificity 66%]). Fair predictor of membranous LN (AUC = 0.75 [sensitivity 75%, specificity 48%]). Good predictor of BCI (AUC = 0.83 [sensitivity 73%, specificity 67%]) Excellent predictor of NIH-AI active LN (AUC = 0.92 [sensitivity 90%, specificity 86%]), 0.94 when corrected for chronicity [sensitivity 90%, specificity 90%]) and good for predicting TIAI active LN (AUC = 0.80 [sensitivity 80%, specificity 68%], 0.83 when corrected for chronicity [sensitivity 90%, specificity 68%]) Fair predictor of response to therapy at baseline (AUC = 0.72), excellent for predicting response to therapy after 3 months (AUC = 0.92) and good after 6 months (AUC = 0.84). Fair predictor of response to therapy at baseline (AUC = 0.74), excellent for predicting response to therapy after 3 (AUC = 0.92) and 6 (AUC = 0.91) months. Fair predictor of active BILAG LN (AUC = 0.724)

Legend: AGP-1 Alpha-1-acid Glycoprotein, AUC area under curve, BAI Biopsy Activity Index, BCI Biopsy Chronicity Index, BILAG British Isles Lupus Assessment Group, CI Confidence Interval, CP Ceruloplasmin, CrCL Creatinine Clearance, IgG Immunoglobulin G, KIM-1 Kidney Injury Molecule-1, LN Lupus Nephritis, L-PGDS Lipocalin-type Prostaglandin-D Synthetase, MCP-1-protein Monocyte Chemoattractant Protein-1, NIH-AI National Institutes of Health-Activity Index, PCR Protein Creatinine Ratio, RAIL Renal Activity Index in Lupus, RAIL-crea Renal Activity Index in Lupus adjusted for urine creatinine, TIAI Tubulointerstitial Activity Index, TF Transferrin, VCAM-1 Vascular cell adhesion protein 1, WCC White Cell Count.

**Table 5 ijms-22-07619-t005:** Biomarkers for other organ involvement.

Biomarker	Study	Findings/Clinical Associations
**CVR biomarkers**		
CIMT CIMT, PWV Lymphopenia CRP hsCRP ApoB:ApoA1 ratio Adiponectin	[114] [115] [116] [116,117] [118] [119] [119]	Higher CIMT was detected in JSLE patients compared to matched HCs (0.45 vs. 0.37 mm, respectively, *p* < 0.0001) Increased CIMT (in 48% vs. 17%, *p* value N/A) and PWV (*p* 0.011) in JSLE patients compared to age and sex-matched HCs Lymphopenia at baseline and at diagnosis were consistently associated with progression of CIMT (*p* = 0.012 and *p* = 0.045, respectively). CRP at baseline positively associated with CTIMP (*p* = 0.049) hsCRP was able to differentiate between matched HCs, and JSLE patients clustered in two groups: with the best and the worst metabolic profiles (defined by a combination of homocysteine, folate, hsCRP, TNF-alpha parameters) (*p* < 0.05 for all comparisons) Strong predictor of CVR (defined by a metabolic signature cross-validated in an adult SLE cohort with atherosclerosis plaques on vascular ultrasound) in JSLE when accounted for BMI (specificity 96.2% and sensitivity 96.7%). Adiponectin correlated with total cholesterol for both JSLE males (r = 0.296; *p* = 0.01) and females patients (r = 0.554; *p* = 0.008); with HDL in females (r = 0.700; *p* = 0.0009) but not males (*p* = 0.32); with LDL in males (r −0.288; *p* = 0.01) but not in females (*p* = 0.22); and with lipoprotein A for males (r = 0.862; *p* < 0.001) but not for females (*p* = 0.18)
**CNS manifestations**		
APLA Anti-ganglioside M1 antibody Anti-aquaporin 4 antibodies Anti-dsDNA, Anti-U1RNP, and Anti-Sm antibodies	[120] [121] [122] [36]	Anticardiolipin antibodies were seen more frequently in children with NPSLE as compared to those without NPSLE (57.8 vs. 23%), lupus anticoagulant was more frequent in children without NPSLE (53.8 vs. 34.7%) (but *p* > 0.05) Significant positive association between anti-ganglioside M1 seropositivity and cognitive dysfunction (*p* < 0.001), as well as with a significant risk for association with cognitive dysfunction (odds ratio: 36; 95% CI: 4.3–302.8) Increased likelihood to experience neurological symptoms (*p* 0.002) and to have received anti-epileptic (*p* = 0.023) and anti-coagulant (*p* = 0.007) Predicted a group of JSLE patients (clustered based on the analysis of the most common autoantibodies found in JSLE in general) with highest prevalence of neuropsychiatric disease (*p* = 0.036)
**Skin manifestations**		
Anti-ds-DNA antibodies	[36]	Increased prevalence of JSLE malar rash in one group of JSLE patients clustered based on frequency of various JSLE autoantibodies (85.3% compared to the other two clusters (53.3%; *p* < 0.001) or cluster 3 (70.7%; *p* = 0.046))
**Haematological manifestations**		
Anti-Ro and anti-ribosomal P antibodies	[36]	Increased prevalence of haemolytic anaemia in one group of JSLE patients clustered based on frequency of various JSLE autoantibodies (40%) compared to the other two clusters (17.7%; *p* = 0.018 and 15.5%; *p* = 0.011, respectively).

Legend: Anti-dsDNA Anti-double stranded Deoxyribonucleic Acid, Anti-Sm Anti-Smith antibodies, Anti-U1RNP Anti-U1 ribonucleoprotein, APLA Antiphospholipid antibody, ApoB:ApoA1 ratio Apolipoprotein B/Apolipoprotein A1 ratio, CNS Central Nervous System, CIMT Carotid intima-media thickness, CRP C-Reactive Protein, CVR Cardiovascular risk, HCs Healthy Controls, HDL high-density lipoprotein, JSLE Juvenile Systemic Lupus Erythematous, LDL low-density lipoprotein NSPLE Neuropsychiatric Lupus Erythematous, PWV Pulse wave velocity, TNF-alpha tumour necrosis factor alpha.

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
