# Peer review of "Biomarkers Associated with Organ-Specific Involvement in Juvenile Systemic Lupus Erythematosus"

_ijms, 2021, doi:10.3390/ijms22147619_

Round 1

Reviewer 1 Report

The aim of this extensive narrative review is to summarize the clinical utility of various traditional and novel biomarkers that have shown a promising role in identifying and predicting specific organ-involvement in JSLE.

I think the review focus on a disease of interest, JSLE, an autoimmune disease at an early age with important organ-complications. I think the review is well organized and comprehensively described and the literature search strategy and studies selection is adequate. However, all the tables included in the review only contemplate biomarkers to renal injury, when in the abstracts mentioning “renal, cardio-vascular, haematological, neurological, skin and joint disease-related JSLE biomarkers”. Thus, title could might seem too generalist for the information finally summarized.

Another consideration, in the review did not search for or analyze important metrics of diagnostic accuracy – sensitivity, specificity, PPV, NPV, c-statistic, etc. Perhaps, because studies do not have them, although shows AUC values. I think that including these parameters in the tables would increase review scientific value.

On the other hand, most of the references to related and previous work showed in this review are not very current, and this fact catches my attention. Overall in tables 2 and 3. It makes me think that perhaps the search strategy is too restrictive or the topic to be discussed is not current, which I don't think is the case.

As minor comments, references and tables format do not follow IJMS template recommendations. Authors should adjust them.

Author Response

The aim of this extensive narrative review is to summarize the clinical utility of various traditional and novel biomarkers that have shown a promising role in identifying and predicting specific organ-involvement in JSLE.

I think the review focus on a disease of interest, JSLE, an autoimmune disease at an early age with important organ-complications. I think the review is well organized and comprehensively described and the literature search strategy and studies selection is adequate.

However, all the tables included in the review only contemplate biomarkers to renal injury, when in the abstracts mentioning “renal, cardio-vascular, haematological, neurological, skin and joint disease-related JSLE biomarkers”. Thus, title could might seem too generalist for the information finally summarized.

We agree with these comments. Because of the paucity of validated biomarker data related to other JSLE organ-manifestations and heterogeneity of data available in the literature, it was not possible to capture the other papers in similarly detailed tables, but we included another table (Table 5) in the manuscript to capture these data, as suggested. We also amended the abstract and introduction to reflect the disparity between the quality of data in JSLE nephritis and other types of organ-involvement.

Another consideration, in the review did not search for or analyze important metrics of diagnostic accuracy – sensitivity, specificity, PPV, NPV, c-statistic, etc. Perhaps, because studies do not have them, although shows AUC values. I think that including these parameters in the tables would increase review scientific value.

The studies that we analysed predominantly provided p values, r values and AUC values. As a general rule, we aimed to include all available statistical data provided by studies in our review. However, where AUC values were given, other metrics such as sensitivity, specificity, NPV, PPV were omitted. We agree with the comments and have re-analysed the studies where AUC values were given. Where further data metrics are available we have now included these in our paper (tracked changes within the manuscript as well as tables).

On the other hand, most of the references to related and previous work showed in this review are not very current, and this fact catches my attention. Overall in tables 2 and 3. It makes me think that perhaps the search strategy is too restrictive or the topic to be discussed is not current, which I don't think is the case.

The scope of this paper as mentioned in the introduction was to highlight the relevance of various JSLE biomarkers, focusing in particular on those which have been tested and validated in JSLE with various organ involvement. Upon repeated literature searches, we have not identified any new papers focused on validated JSLE biomarkers from 2019-2021 to add to Table 2. However, we identified one additional paper on a urinary biomarker (urinary sCD25) and one additional paper on a serum biomarker (serum NK cells) which investigated their role as biomarkers of LN. These have been added to the manuscript text as well as in Table 3. We also identified a cluster analysis paper looking at the value of combinations of autoantibodies in predicting clusters of clinical manifestations which has also been added to the manuscript and tables (Jurencak et al., 2009).

However, if there are any key references focused on validated JSLE biomarkers published more recently that our second searches have missed, we will be grateful for the suggestion of adding them to our tables.

In an effort to update our references, we also cited in text more recent papers, such as:

A panel of urinary proteins predicts active lupus nephritis and response to rituximab treatment which unfortunately does not include any JSLE patients.

Our literature search identified a few recent papers which have investigated organ biomarkers in adult SLE, however these were not included as they were beyond the scope of the paper.

Additionally our literature search identified a number of recent papers which investigated novel biomarkers as a marker of general (not organ specific) JSLE activity, which we also included for completion.  

  1. Microarray expression profile of circular RNAs and mRNAs in children with systemic lupus erythematosus
  2. Elevated serum interleukin-34 level in juvenile systemic lupus erythematosus and disease activity
  3. SIGLEC1 (CD169) is a sensitive biomarker for the deterioration of the clinical course in childhood systemic lupus erythematosus

As minor comments, references and tables format do not follow IJMS template recommendations. Authors should adjust them.

The references and table format have been amended accordingly.

Reviewer 2 Report

In this review, the authors address most of the works on the biomarkers used in the management of jSLE. There is a number of recent reviews on sJLE (i.e., PMID: 33569643, 32725543), although in this manuscript authors bring new evidences on new biomarkers (they are the only new data in the reference list. Otherwise, they are old references) and they could have some interest.

My main concerns is about the order the authors explain biomarkers. Thus, IFN-I related and innate immune system biomarkers are explained in different parts of the text according to the organ related in jSLE. The same is for autoantibodies. Moreover, they consider autoantibodies as simple biomarkers when they have much to do in jSLE. Not only that, but they use dsDNA for anti-dsDNA antibodies and the only explain double strand DNA in the abbreviation.

From my point of view, the explanation of autoantibodies is quite important and merits one single heading in the review, addressing their role in the different manifestations of SLE. Furthermore, some additional autoantibodies (not only anti-dsDNA in nephritis or ACPA in arthritis) should be included, such as anti-nuclear antibodies as a whole, anti-chromatine, anti-C1q, etc. Besides, some references to the methodological aspects of the differences between studies could be discussed. This is the same for complement.

In general, there are some parts of the text that go deeper in to pathophisiological aspects but other remain only in the clinical aspects (i.e., IFN-type I response or miRNA).

Author Response

In this review, the authors address most of the works on the biomarkers used in the management of jSLE. There is a number of recent reviews on sJLE (i.e., PMID: 33569643, 32725543), although in this manuscript authors bring new evidences on new biomarkers (they are the only new data in the reference list. Otherwise, they are old references) and they could have some interest.

My main concerns is about the order the authors explain biomarkers. Thus, IFN-I related and innate immune system biomarkers are explained in different parts of the text according to the organ related in jSLE. The same is for autoantibodies. Moreover, they consider autoantibodies as simple biomarkers when they have much to do in jSLE. Not only that, but they use dsDNA for anti-dsDNA antibodies and the only explain double strand DNA in the abbreviation.

From my point of view, the explanation of autoantibodies is quite important and merits one single heading in the review, addressing their role in the different manifestations of SLE. Furthermore, some additional autoantibodies (not only anti-dsDNA in nephritis or ACPA in arthritis) should be included, such as anti-nuclear antibodies as a whole, anti-chromatine, anti-C1q, etc. Besides, some references to the methodological aspects of the differences between studies could be discussed. This is the same for complement.

In general, there are some parts of the text that go deeper in to pathophisiological aspects but other remain only in the clinical aspects (i.e., IFN-type I response or miRNA).

Thank you for your comments. The authors kept the structure of the review in concordance with the title focusing mainly on organ-specific JSLE biomarkers rather than JSLE biomarkers in general.

This is particularly relevant for the inter-disciplinary therapeutic approach and research in JSLE. In addition, there are many literature reviews focused on lupus biomarkers in general (particularly focused on autoantibodies, which unfortunately have quite poor predictor value as biomarkers as very limited clinical utility) or renal lupus biomarkers in particular; therefore, the authors appreciated that a different review format and scope was warranted.

We appreciated the value of this suggestion, and revised the manuscript and included (with tracked changes):

  1. a large general paragraph focused on the role autoantibodies in JSLE overall
  2. expanded on specific autoantibodies under the relevant organ-specific sections (although in keeping with the scope of this review, which is not meant to be exhaustive)
  3. included relevant organ-specific auto-antibody biomarkers in a new table (Table 5)
  4. discussed differences in the methodological aspects of various assays for detection of autoantibodies in the discussion
  5. expanded on the pathophysiological aspects of type IFN response, miRNA

Round 2

Reviewer 2 Report

Authors have addressed most of my concerns and questions. However, and despite they introduced a section about autoantibodies in general, they indicate the role of autoantibodies in each organ involvement and not only, since they mention the role of C3 and C4 in LN, what has to do undoubtely with antibodies, probably anti-dsDNA.

Minor changes:

  • Reference 24 about classification criteria for SLE. There are more recent references about new criteria by ACR and EULAR.
  • There is an important mistake in the text (third paragraph in page 11 and in table). NK cells are not measured in serum but in whole blood. They are peripheral blood or circulating NK cells. Please, correct elsewhere.

Author Response

Thank you for your suggestions. They have all been addressed and an updated version of the paper has been submitted.